# Adversarial Dual On-Policy Distillation from Expressive Teacher

**Zhenglin Wan** [* 1]  **Jingxuan Wu** [* 2]  **Xingrui Yu** [3 4]  **Chubin Zhang** [5]  **Mingcong Lei** [6]  **Bo An** [7]  **Ivor W. Tsang** [3 4 7]  **Yang You** [1]

## Abstract

Learning from demonstrations in embodied control is often cast as behavioral cloning, and recent diffusion or flow-matching policies improve this paradigm by modeling multi-modal expert actions. Yet these methods remain offline supervised learners: the policy is trained only on expert states and receives no corrective signal on the states it actually visits. On-policy distillation (OPD) offers a natural remedy, but standard OPD assumes a strong fixed teacher, which is unavailable in demonstration-only control. We propose **FA-OPD**, an *adversarial dual on-policy distillation* method in which a Flow Matching (FM) teacher is learned from demonstrations and co-trained with a lightweight MLP student. The teacher provides two complementary signals on student rollouts. The reward channel learns an expert-likeness objective over state-action pairs and drives online exploration through long-horizon policy optimization. The action channel supplies dense local targets at student-visited states, stabilizing exploitation. FA-OPD couples them so that reward distillation enables generalization beyond point-wise demonstrations, while action distillation keeps exploration anchored near expert-like behavior. Across six robot navigation, manipulation, and locomotion benchmarks, FA-OPD beats strong baselines and shows much stronger robustness under noisy or limited demonstrations. Source code: https://github.com/vanzll/FA-OPD.

---

[1]School of Computing, National University of Singapore [2]Department of Statistics and Operations Research, UNC-Chapel Hill, America [3]CFAR, Agency for Science, Technology and Research, Singapore [4]IHPC, Agency for Science, Technology and Research, Singapore [5]Beijing University of Posts and Telecommunications, China [6]School of Data Science, The Chinese University of Hong Kong, Shenzhen, China [7]College of Computing and Data Science, Nanyang Technological University, Singapore. Correspondence to: Xingrui Yu, Yang You <yu_xingrui@a-star.edu.sg, yangyou@nus.edu.sg>.

*Proceedings of the 43rd International Conference on Machine Learning*, Seoul, South Korea. PMLR 306, 2026. Copyright 2026 by the author(s).

## 1. Introduction

Learning from demonstrations (LfD) is a central setting in robot navigation, manipulation, and locomotion, where specifying a reward is difficult but collecting demonstrations is feasible. The most direct approach is behavioral cloning (BC): train a policy by supervised learning on expert state-action pairs. Recent diffusion and Flow Matching (FM) policies have made this supervised paradigm substantially more expressive by fitting complex, multi-modal action distributions that simple Gaussian heads cannot represent (Chi et al., 2024; Zhang et al., 2025a;b; Braun et al., 2024). Flow Matching is especially attractive because it offers efficient training and stable sampling for multi-modal data (Lipman et al., 2022; Liu et al., 2025).

However, the limitation is that BC remains an offline supervised learner. The policy is trained on expert states but deployed on its own induced state distribution, so early mistakes can move it outside the demonstration manifold and compound over time (Ross et al., 2011; Simchowitz et al., 2025). This exposure-bias problem becomes sharper when demonstrations are limited, noisy, or sub-optimal (Yu et al., 2025; Wan et al., 2025c; 2024), as we observe in Section 4.3. Expressive diffusion or FM policy classes improve how well the expert dataset is fit, but they do not by themselves provide corrective supervision on the states the learned policy actually visits.

On the other hand, *on-policy distillation* (OPD) (Agarwal et al., 2023; 2024; Lu, 2025; DeepSeek-AI, 2026) gives a useful template for this problem: let the student roll out, query a teacher for dense supervision on the student's own distribution, and update the student from that feedback. Existing OPD methods usually assume that the teacher is already a strong fixed model. This assumption is natural in settings such as language-model post-training, but it is often unavailable in embodied tasks. In our setting, the only supervision is a static demonstration set. Thus the teacher must itself be learned from the demonstrations and remain calibrated as the student distribution changes. We therefore ask: *can OPD work for embodied tasks when the teacher is learned and co-trained with the student?*

Recent language-model post-training systems make this motivation concrete: Thinking Machines Lab frames OPD as

the combination of on-policy student rollouts with dense teacher grading for RL training, while DeepSeek-V4 consolidates multiple domain specialists through on-policy reverse-KL distillation and full-vocabulary logit matching (Lu, 2025; DeepSeek-AI, 2026). These examples suggest two broad supervision styles: score-based feedback, where the teacher signal is optimized through an RL-style objective, and target-based feedback, where teacher outputs are treated as supervised targets through KL-style distillation (Agarwal et al., 2023; 2024). Translating this split to embodied control gives the central design question of FA-OPD: *what should the learned teacher provide?* A scalar score and an action target have complementary roles. The score is the exploration-driving channel: by learning an adversarial expert-likeness objective over state-action pairs, it can evaluate the student's own visits, including states not exactly present in the demonstrations, and supports long-horizon policy improvement through PPO. However, this learned objective can extrapolate poorly if the policy drifts far from demonstration support. The action target is the exploitation-stabilizing channel: it gives dense, low-variance local corrections at student-visited states, but by itself it reduces to on-policy imitation without an explicit scalar objective for online improvement. A useful embodied OPD teacher should therefore provide both signals: the reward channel drives online improvement, while the action channel keeps that reward-guided exploration anchored near expert-like behavior.

To this end, we propose **FA-OPD**, an adversarial dual on-policy distillation method built around an adversarially co-trained Flow Matching teacher. On each rollout, the same FM teacher supplies two channels of supervision: (i) it *scores* each $(s, a)$ the student visits via an adversarial FM-enhanced discriminator (*reward distillation*, used as a reward in RL), and (ii) it *shows* expert-like target actions at student-visited states for the student to regress onto via MSE (*action distillation*, the on-policy analog of DAgger (Ross et al., 2011)). The teacher is learned and co-trained as a binary classifier between expert and student visits, so the supervision signal sharpens where the student currently needs it. We defer the analysis of why FM is a good teacher choice (its loss is an ELBO-based proxy for the optimal scoring objective) to Appendix D.

This design uses FM for what it is good at–distribution-aware supervision–while avoiding the main practical costs of deploying an FM policy directly. FM action generation requires iterative sampling or ODE/flow inference, and direct online policy-gradient updates must either differentiate through those iterations or approximate the needed density/importance ratios, which is unstable and costly in practice (Park et al., 2025; Zhang et al., 2026) (Appendix C). In FA-OPD, the FM teacher supervises training, but gradients flow only through a Gaussian MLP student; policy optimization

stays within a standard PPO loop, and deployment is a single MLP forward pass. We also discuss why a unimodal Gaussian student can still benefit from a multi-modal teacher in Appendix A.

We summarize our contributions:

- We introduce **adversarial dual on-policy distillation**, a framework that couples a scalar reward channel for online improvement with an action-target channel for local correction. Our ablations show neither channel alone achieves comparable performance; their complementarity under a co-trained teacher is the core architectural contribution.

- We instantiate this framework with a Flow Matching teacher, giving **FA-OPD**. The framework does not depend on this choice; we explain in Appendix D why FM is a good fit.

- Across six robot navigation, manipulation, and locomotion benchmarks, FA-OPD beats strong baselines, is much more robust to noisy or limited supervision and to OOD states, and runs at MLP-policy speed at deployment.

## 2. Background

### 2.1. Flow Matching (FM) for Generative Modeling

Following the generative modeling paradigm, FM aims to learn a generator $G_\theta$ that produces samples aligning with a target distribution, based on samples from this distribution. FM assumes the existence of a continuous flow that transforms a simple initial distribution $\mathcal{N}(0, I)$ at $t = 0$ to the target distribution at $t = 1$. This transformation is defined by the ordinary differential equation:

$$\frac{d}{dt}\phi_t(x) = v_t(\phi_t(x)), \quad \phi_0(x) = x.$$

The core idea is to regress a parametric vector field $v_t(x; \theta)$ toward a target vector field $u_t(x)$ that generates a desired probability density path $p_t(x)$. This is formalized through the FM objective:

$$\mathcal{L}_{\text{FM}}(\theta) = \mathbb{E}_{t, p_t(x)} \|v_t(x) - u_t(x)\|^2,$$

where $t \sim \mathcal{U}[0, 1]$ and $x \sim p_t(x)$. Since the marginal vector field $u_t(x)$ and probability path $p_t(x)$ are generally intractable, Conditional Flow Matching (CFM) (Lipman et al., 2022) provides a practical alternative. Let $q(x_1)$ be the data distribution and $p_t(x|x_1)$ be a conditional probability path satisfying $p_0(x|x_1) = p(x)$ and $p_1(x|x_1) = \mathcal{N}(x_1, \sigma_{\min}^2 I)$. The marginal path is:

$$p_t(x) = \int p_t(x|x_1)q(x_1)dx_1 = \mathbb{E}_{x_1}[p_t(x|x_1)].$$

The CFM objective is defined as:

$$\mathcal{L}_{\text{CFM}}(\theta) = \mathbb{E}_{t,q(x_1),p_t(x|x_1)} \|v_t(x) - u_t(x|x_1)\|^2.$$

Crucially, it is shown that $\nabla_\theta \mathcal{L}_{\text{FM}}(\theta) = \nabla_\theta \mathcal{L}_{\text{CFM}}(\theta)$, making CFM a tractable training objective.

A common choice for the conditional path is a Gaussian parameterization: $p_t(x|x_1) = \mathcal{N}\left(x \mid \mu_t(x_1), \sigma_t(x_1)^2 I\right)$, with boundary conditions $\mu_0(x_1) = 0$, $\sigma_0(x_1) = 1$, $\mu_1(x_1) = x_1$, and $\sigma_1(x_1) = \sigma_{\min}$. The conditional path is manually defined (e.g., function $u_t(x_1)$ and $\sigma_t(x_1)$).

## 2.2. Inverse Reinforcement Learning and Adversarial Imitation Learning

Inverse Reinforcement Learning derives a reward function given expert demonstrations, and optimize the policy based on this reward function (Ng et al., 2000). Belonging to IRL paradigm, adversarial imitation learning (AIL) reframes the problem through adversarial training. It employs a discriminator $D(s, a)$ to distinguish between expert and agent trajectories, while the policy $\pi$ is optimized to generate trajectories that fool the discriminator. The discriminator is optimized through the following objective:

$$\max_D \quad \mathbb{E}_{(s,a)\sim\rho_E}[\log D(s, a)] + \mathbb{E}_{(s,a)\sim\rho_\pi}[\log(1 - D(s, a))],$$

where $\rho_E$ and $\rho_\pi$ represent the state-action distributions of the expert and the learning agent respectively. The discriminator's output provides an adaptive reward signal that guides the policy optimization. Although the discriminator-derived reward in our method is grounded in the AIL paradigm, we refer to the overall approach as FA-OPD because it complements this reward-based distillation with a second, action-based distillation term (Sec. 3.4) within a unified on-policy distillation framework.

## 2.3. On-Policy Distillation

On-policy distillation (OPD) (Agarwal et al., 2023; 2024) trains a *student* policy by querying a fixed *teacher* on outputs the student itself generates. Formally, the student rolls out $(s, a) \sim \pi_\phi$, the teacher returns a per-sample score $s_T(s, a)$ (e.g., teacher log-probability or a reward-model score), and the student is updated to maximize $\mathbb{E}_{(s,a)\sim\pi_\phi}[s_T(s, a)]$ via RL or REINFORCE-style gradients. OPD addresses the well-known exposure-bias problem of off-policy supervised distillation: the student is supervised on the states it actually visits, so it never learns to recover from its own mistakes off-distribution. In existing OPD methods the teacher is assumed pre-trained and frozen (e.g., a larger LLM, a value model, or a reward model fitted offline). The setting of this paper differs in that no such pre-trained teacher exists; the teacher must be *learned from* the same expert demonstrations the student is trying to imitate, and we will see that it is

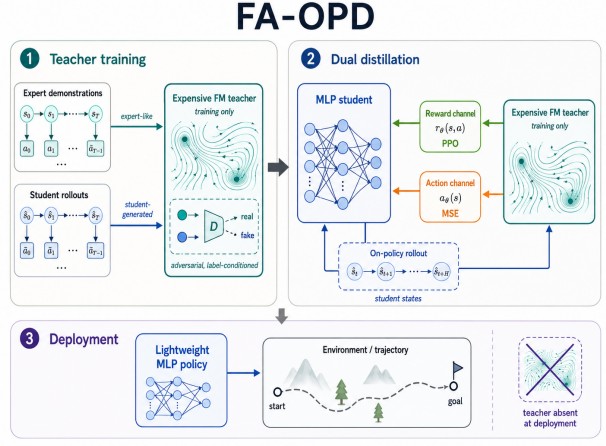

*Figure 1.* FA-OPD uses the FM teacher only during training. It distills to the MLP student through two channels: reward distillation supplies expert-likeness scores for PPO, while action distillation supplies dense target actions at student-visited states. Only the MLP student is deployed.

most useful when *co-trained adversarially* with the student rather than pre-trained.

## 3. Methodology

In this work, we propose FA-OPD, a teacher-student architecture where a training-time FM teacher guides the online update of a lightweight MLP student from expert demonstrations. Figure 1 illustrates the general workflow of FA-OPD. We first explain why FM is useful as an expressive teacher but unattractive as the deployed online actor, then introduce the FM-enhanced discriminator for reward distillation, and finally present action distillation and the combined student objective.

### 3.1. Flow Matching as an Expressive Policy Class

We begin with how flow matching could be utilized to model the policy in RL. Flow Matching is known for its strong capability to fit complex data distributions, which is particularly essential in robot learning. This is because real-world demonstrations often exhibit multi-modal characteristics arising from diverse expert behaviors (Black et al., 2024). The FM policy is formulated via a conditional FM model defined by an ordinary differential equation (ODE):

$$\pi_\theta(a \mid s) = p_\theta(a_1 \mid s) = \int p_\theta(a_{0:T} \mid s)da_{0:T-1}, \quad (1)$$

where $p_\theta$ means the probability density function corresponding to specific vector field parametrized by $\theta$, $T$ means the number of discrete time steps used to approximate the continuous probability flow, and $a_t$ denotes the status at flow time $t \in [0, 1]$, with $a_0 \sim \mathcal{N}(0, I)$ as the prior distribution

and $a_1$ as the action used for the policy. The probability path is modeled through a vector field $v_\theta$ conditioning on $s$:

$$\frac{da_t}{dt} = v_\theta(a_t, s, t). \qquad (2)$$

To train the FM policy, the most common approach is behavioral cloning (Torabi et al., 2018) based on the Flow Matching loss (Lipman et al., 2023):

$$\mathcal{L}_{\text{FM}}(\theta) = \mathbb{E}_{t\sim\mathcal{U}[0,1],(s,a_1)\sim\mathcal{D}_E,a_t\sim p_t(a|s,a_1)}\Big[ \\ \|v_\theta(a_t, s, t) - u_t(a_t \mid a_1, s)\|^2 \Big] \qquad (3)$$

where $\mathcal{D}_E$ is the expert dataset, $u_t(a_t \mid a_1, s)$ is the pre-defined conditional flow velocity, and $a_t$ is sampled from the pre-defined conditional probability path connecting $a_0$ and $a_1$ given $s$. Upon sufficient training of the FM policy, actions are generated by solving the ODE forward in time:

$$a_1 = a_0 + \int_0^1 v_\theta(a_t, s, t)dt, \qquad (4)$$

using numerical ODE solvers.

An FM policy is attractive for cloning multi-modal expert behavior, but offline cloning still lacks corrective interaction with the environment and generalizes poorly when demonstrations are limited or sub-optimal. Directly making the FM policy the online RL actor is also unattractive: each action requires iterative flow inference, and policy-gradient updates must backpropagate through this process or approximate intractable density/importance ratios. This makes the FM policy gradient **hard to compute** and costly in practice; Appendix C provides theoretical details.

### 3.2. FM-Enhanced Discriminator

To address these limitations, FA-OPD keeps FM as a training-time teacher and deploys a cheap PPO-compatible MLP student. Motivated by Adversarial Imitation Learning (AIL) (Ho & Ermon, 2016; Peng et al., 2018; Wang et al., 2024; Lai et al., 2024), which learns a discriminator to distinguish expert and agent behavior and provides reward signals for online policy update, we use an FM teacher to model the expert state-action joint distribution and transfer this distributional knowledge into an AIL reward. Specifically, we replace the traditional MLP-based discriminator in Eq. 5 by an FM-enhanced discriminator $D_{\text{FM}}$, and update the discriminator via:

$$\min_\theta \quad \mathcal{L}_{\text{FM}} = \mathbb{E}_{(s,a)\sim\rho_E}[\log(1 - D_{\text{FM},\theta}(s,a))] + \\ \mathbb{E}_{(s,a)\sim\rho_\pi}[\log(D_{\text{FM},\theta}(s,a))]. \qquad (5)$$

Ideally, the FM-enhanced discriminator could take a state-action pair as input and output a scalar score answering:

"*how likely is this action $a$ under the expert data distribution at state $s$?*" The traditional AIL discriminator adopts an MLP architecture and naively models the data-point-level similarity (Wang et al., 2024). In contrast, the FM-enhanced discriminator models the distribution-level similarity between the agent's state-action pair and expert data distribution. In this way, the FM teacher's distributional knowledge is distilled into the discriminator output, which is used to shape the reward signal:

$$r_\theta(s, a) = \log(D_{\text{FM},\theta}(s,a)) - \log(1 - D_{\text{FM},\theta}(s,a)). \qquad (6)$$

This framework fully follows the paradigm of AIL, and the reward function is adapted from AIRL (Fu et al., 2017), a popular variant of AIL, to provide a dense and guiding reward signal that encourages the agent's policy to match the expert's distribution during online update.

**OPD interpretation.** The construction above is equivalently an on-policy distillation of the FM teacher into the MLP student: at each visit $(s, a) \sim \pi_\phi$, the teacher returns a per-action score (the AIRL log-ratio computed from $D_{\text{FM},\theta}$, Eq. 6), and the student is updated to maximize that score via PPO. The novelty over standard OPD lies in the teacher itself: instead of being a frozen pre-trained model, it is co-trained adversarially against the student via Eq. 5, so the score becomes a better-calibrated indicator of expert-likeness as training progresses (we formalize this in Appendix D).

### 3.3. How to design $D_{\text{FM},\theta}$?

The design philosophy of FM-enhanced discriminator $D_{\text{FM},\theta}$ is essential in our work since it directly affects the quality of reward signals in terms of the expressiveness of expert behavior. Inspired by FPO (McAllister et al., 2025) which suggests that the loss value of an FM policy is a strong positive indicator of the similarity between one specific input and the target distribution, we adapt this insight into the design of $D_{\text{FM},\theta}$. Specifically, the "teacher" FM model takes the joint $(s, a)$ pair as input during training, modeling not only the action pattern but also the state distribution (the reason for such design will be explained in Appendix A). Then, the loss function is:

$$\mathcal{L}_{\text{FM}}(\theta) = \mathbb{E}_{t\sim\mathcal{U}[0,1],(s_1,a_1)\sim\mathcal{D}_E,(s_t,a_t)\sim p_t(\cdot|(s_1,a_1))} \\ [\|v_\theta((s_t, a_t), t) - u_t((s_t, a_t) \mid (s_1, a_1))\|^2]. \qquad (7)$$

Intuitively, the loss value will decrease as the FM model gradually fits the distribution of dataset $D$. Therefore, we can use the loss value to measure the distribution-level distance between one specific data point $(s', a')$ and the target distribution by canceling the uncertainty of the expectation that comes from $(s_1, a_1) \sim \mathcal{D}$ and replaces the $(s_1, a_1)$ by

$(s', a')$ (McAllister et al., 2025):

$$\text{Dist}_\theta(s', a') = \mathbb{E}_{t \sim \mathcal{U}[0,1], (s_t, a_t) \sim p_t(\cdot | (s', a'))}$$
$$[\|v_\theta((s_t, a_t), t) - u_t((s_t, a_t) \mid (s', a'))\|^2]. \quad (8)$$

Meanwhile, inspired by (Lai et al., 2024) and to help the discriminator to discern the expert data and agent data, we further improve the design by conditioning the "teacher" FM model on an indicator variable $c$, which takes value from $\{0, 1\}$ to represent whether the FM model is fitting expert data or agent data:

$$\text{Dist}_\theta(s', a' | c) = \mathbb{E}_{t \sim \mathcal{U}[0,1], (s_t, a_t) \sim p_t(\cdot | (s', a'))}$$
$$[\|v_\theta((s_t, a_t), t | c) - u_t((s_t, a_t) \mid (s', a'), c)\|^2], \quad (9)$$

where $\text{Dist}(s', a' | c = 1)$ represents the distance between $(s', a')$ and expert data distribution, $\text{Dist}(s', a' | c = 0)$ means the distance between $(s', a')$ and agent data distribution. Then, the FM-enhanced discriminator $D_{\text{FM},\theta}(s, a)$ is given by a temperature-scaled Softmax transformation:

$$\frac{\exp\big(-\text{Dist}_\theta(s, a \mid c{=}1)/\tau\big)}{\exp\big(-\text{Dist}_\theta(s, a \mid c{=}1)/\tau\big) + \exp\big(-\text{Dist}_\theta(s, a \mid c{=}0)/\tau\big)}. \quad (10)$$

The temperature $\tau > 0$ controls the sharpness of the discriminator. The raw Dist values are not bounded inside $[0, 1]$: in our experiments they fluctuate from sub-unit values for well-fit expert points up to magnitudes above 1 for out-of-distribution agent rollouts during early training, depending on the environment dimensionality. Without scaling, this would push the Softmax into hard $\{0, 1\}$ regimes and destabilize the AIRL log-ratio reward (Eq. 6). We adopt a single fixed $\tau = 0.1$ for all environments without per-task tuning; the role and chosen value are reported in Appendix E (Table 2). This formulation keeps the output normalized within $[0, 1]$, makes it compatible with the traditional AIL setting of Eq. 5, and yields a calibrated indicator of distribution-level similarity between $(s, a)$ and the expert data while remaining aware of the agent's data.

### 3.4. Action Distillation

The reward distillation of Sec. 3.2–3.3 supervises the student through a single scalar score per visit, which PPO can optimize over rollouts but can be noisy on out-of-distribution states where the FM-enhanced discriminator may extrapolate poorly (Fujimoto et al., 2019). We complement it with a second, denser form of supervision: at each state $s$ the student visits during its on-policy rollouts, we let the FM teacher generate an expert-distributed action $a_G$ and ask the student to regress onto it. This is the on-policy analog of DAgger (Ross et al., 2011): the student is corrected at exactly the states it actually encounters, using teacher-supplied targets, with the key difference that our teacher is not an

oracle expert but a flow-matching generator co-trained with the student. The combined objective is:

$$\max_\phi \quad \mathcal{J}(\phi) = \underbrace{\mathbb{E}_{(s,a) \sim \pi_\phi} \left[ \sum_k \gamma^k r_\theta(s, a) \right]}_{\text{reward distillation (Sec. 3.2)}} -$$
$$\beta \underbrace{\mathbb{E}_{\substack{s \sim \rho_{\pi_\phi}, a_\pi \sim \pi_\phi(\cdot|s) \\ a_G \sim G_\theta(\cdot|s, c=1)}} \left[ \|a_\pi - a_G\|^2 \right]}_{\text{action distillation (DAgger-style)}}, \quad (11)$$

where $\rho_{\pi_\phi}$ is the student's state visitation distribution, $G_\theta$ is the FM generator role of the teacher (sharing parameters with the FM-enhanced discriminator), $\gamma$ is the discount factor, and $\beta$ trades off the two distillation modes. The student policy $\pi_\phi$ is a simple MLP throughout, so gradients of both terms flow only through an affine reparameterization (avoiding the BPTT pathology of FM-policy methods, Appendix C).

**Why dual distillation?** FA-OPD combines two forms of on-policy distillation that supervise the student on its own rollouts but use the shared FM teacher in different roles. In *reward distillation* (Sec. 3.2, Sec. 3.3), the teacher acts as an FM-enhanced scorer: it scores each visited $(s, a)$, and PPO uses this score as a per-action reward. This channel supplies an expert-likeness objective for exploration and long-horizon online improvement. In *action distillation*, the same teacher acts as an FM generator: at each student-visited state, it generates an expert-like target action, and the student is regressed onto it via MSE. This channel supplies dense local correction for exploitation, in the spirit of DAgger (Ross et al., 2011), except our teacher is a learned, co-trained generator rather than an oracle expert.

Each mode alone has a known failure case. With reward only, the student can explore and optimize a learned expert-likeness objective, but the learned reward may extrapolate poorly if the policy drifts far from demonstration support. With action only, the student receives stable local corrections, but the objective becomes imitation-like and does not provide a scalar signal for reward-guided exploration. The dual objective makes the two channels regulate each other: reward distillation gives the student an online improvement signal, while action distillation converts the FM teacher's local generative knowledge into a stabilizing exploitation signal. Both modes share a single conditional FM teacher, co-trained as a binary classifier between expert and student visits (Sec. 3.3); our ablations (Appendix F.2) show that neither $\beta = 0$ (reward only) nor overly large $\beta$ (action-dominated training) matches the dual setting; the best $\beta$ is in the middle.

# 4. Experiments

**Evaluation protocol.** Although FA-OPD is framed as on-policy distillation, our experiments use the standard *learning-from-demonstrations* (LfD) setup: each task gives a static set of expert trajectories and no true environment reward at training time. This setup is the same as the one in the imitation-learning (IL) literature, so we compare against the strongest IL baselines under this data condition: Diffusion Policy, Flow Matching Policy, the GAIL family, and DRAIL. The comparison is meaningful regardless of how each method frames itself internally.

As illustrated in Figure 5 in Appendix, we evaluate our method across six environments spanning navigation, locomotion, and manipulation. Each single experiment is repeated for 4 random seeds. Appendix E shows the hardware setup and more details. Our experiments aim to answer four research questions (RQs):

- **[RQ 1.]** How efficient is FA-OPD in terms of the convergence speed to learn a high-performing policy?

- **[RQ 2.]** Does FA-OPD really helps the learned policy to generalize to unseen states better?

- **[RQ 3.]** Could FA-OPD show stronger robustness against sub-optimal demonstrations?

- **[RQ 4.]** Why not simply integrate FM into online RL to address the limitation of FM policies?

## 4.1. [RQ 1.] Learning Efficiency Study

First, we evaluate the efficiency and effectiveness with which FA-OPD learns to perform the task from expert demonstrations. For tasks with a binary success indicator—including Ant-goal, Hand-rotate, Fetch-pick, and Maze2d—we report the training curve of success rate (y-axis, range 0–1) against training steps (x-axis). For tasks without a binary outcome measure, namely Hopper and Walker2d, we report the average return (y-axis, range 0 to $+\inf$) over training steps. To assess the learning efficiency of FA-OPD, we compare it with two categories of baselines: (1) Supervised Behavioral Cloning methods: Diffusion Policy (DP) (Chi et al., 2024) and Flow-Matching Policy (FP) (Zhang et al., 2025a); and (2) Inverse Reinforcement Learning (IRL) methods: GAIL (Ho & Ermon, 2016), VAIL (Peng et al., 2018), WAIL (Xiao et al., 2019), AIRL (Fu et al., 2017), and DRAIL (Lai et al., 2024). Note that methods in category (1) do not involve online policy updates, so their performance curves appear as horizontal lines. This comparison helps illustrate how FA-OPD addresses key limitations of FP (as well as DP). Comparisons with category (2) demonstrate the advantages of FA-OPD within the IRL paradigm. Please refer to Appendix G and

G.1 for the technical details of baselines, the discussion about the difference between FA-OPD and DRAIL.

As shown in Figure 2, FA-OPD achieves the best final performance across all six environments except Ant-goal and reaches near-100% success in Hand-rotate and Fetch-pick. Compared to other IRL algorithms, FA-OPD converges faster and attains better ultimate performance in Hand-rotate, Fetch-pick, and Walker2d, owing to its strong capacity to model complex expert distributions and maintain stable policy improvement during online updates. In the remaining three environments, it still outperforms other IRL methods by a smaller margin while exhibiting lower variance, indicating higher stability. Relative to supervised behavioral cloning methods (DP/FP), FA-OPD substantially surpasses them in all environments except Ant-goal, highlighting a fundamental limitation of Flow Matching Policy and Diffusion Policy—the lack of active exploration. FA-OPD addresses this through online reinforcement learning and a balanced exploration–exploitation strategy, enabling robust handling of unseen states. In Ant-goal, FA-OPD performs similarly to DP and FP because the task's simple path to the goal is fully covered by the expert data, making extensive exploration unnecessary. The quantitative results of this study is provided in Table 3 at Appendix F.1.

## 4.2. [RQ 2.] Generalization Study

To assess FA-OPD's ability to generalize to unseen states, we conducted a generalization study on **Manipulation** tasks (Figure 3 and Figure 7 in Appendix). Specifically, we introduced varying levels of noise to the initial and goal states in the Hand-rotate and Fetch-pick environment, testing whether the learned policies of FA-OPD and baseline methods could adapt to new trajectories under perturbed conditions. Noise scales included 1× (original expert setting), 1.25×, 1.5×, 1.75×, 2.00×, and 2.25×, where 1.25× denotes noise 1.25 times greater than that used during expert data collection. As shown in Figure 7, all methods exhibit performance degradation in Hand-rotate as noise increases, yet FA-OPD consistently outperforms the baselines across all noise levels. Specifically, FA-OPD maintains a success rate above 0.95 up to 1.5× noise, while the performance of DP and FP declines more sharply, dropping from 0.91 to nearly 0.82. This underscores the importance of online environmental interaction—even for policies capable of modeling complex action distributions. On the other hand, GAIL shows significant performance degradation (from around 0.91 to nearly 0.5) with only slight additional noise, and WAIL fails extensively under higher noise conditions. The limited generalization ability of GAIL and WAIL highlights the key role that FM played in capturing the multi-modal state-action distribution to handle diverse scenarios.

Figure 3 shows the final performance of each method af-

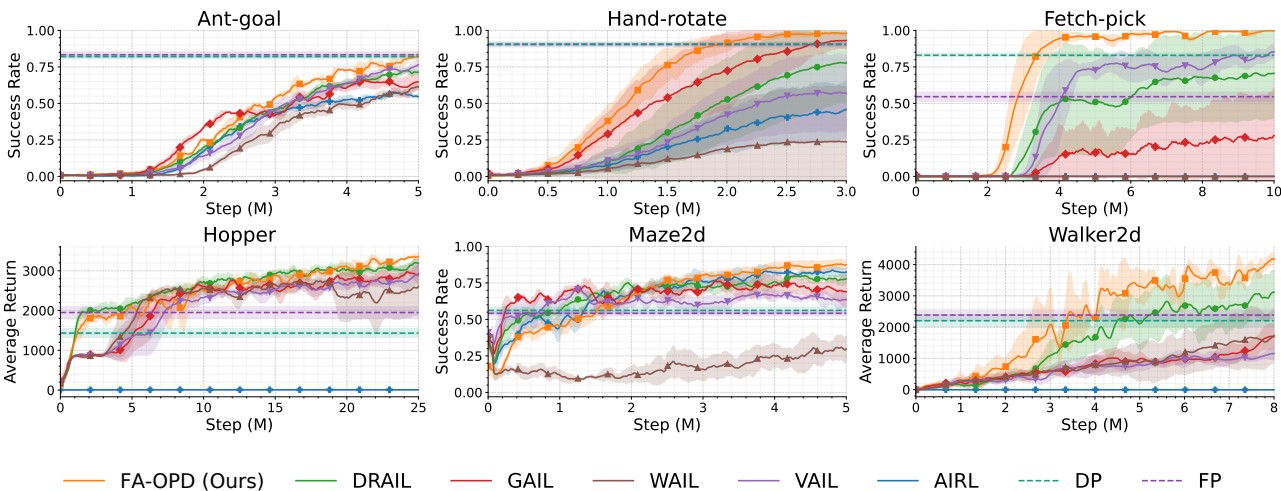

*Figure 2.* Learning curve of FA-OPD and baselines across 6 environments.

ter $10^7$ training steps in the Fetch-pick environment, with varying noise levels defined similarly as in Hand-rotate. We observe similar but more contrasting and compelling results: FA-OPD suffers from minimal and negligible performance degradation as noise levels increase. For baselines, static methods like FP and DP steadily suffer from performance drops due to the lack of generalization in behavior cloning. Notably, a large proportion of baselines fail to learn any patterns from demonstrations (i.e., achieve 0 success rate) as noise levels increase, such as GAIL, WAIL, and AIRL. The most competitive baselines in this environment, namely DRAIL and VAIL, can roughly maintain their performance when noise levels are not large (i.e., from 1.00 to 1.25), but suffer from drastic drops when noise levels further increase. Overall, FA-OPD demonstrates much stronger generalization to unseen states compared to all baselines in Fetch-pick, due to its policy expressiveness during training and the combination of reward and action distillation under online interaction. To further justify our claim, we also conducted the generalization study in **Navigation** and **Locomotion** tasks in Appendix F.3.

### 4.3. [RQ 3.] Robustness Study

*Table 1.* Robustness in Walker2d: We compare Diffusion Policy (DP), Flow-Matching Policy (FP), and our FA-OPD with sub-optimal experts, reporting average return (± std. over four seeds), with the best result highlighted in bold. The higher the ID values, the more optimal the expert data.

| ID | Expert Return | DP | FP | FA-OPD |
|---|---|---|---|---|
| 1 | 265.36 | **247.24** (±19.3) | 218.33 (±20.2) | 99.05 (±5.1) |
| 2 | 1965.58 | 1270.25 (±143.7) | 1342.47 (±168.1) | **2093.14** (±311.7) |
| 3 | 2662.14 | 1710.47 (±154.5) | 1578.91 (±175.6) | **2618.49** (±356.9) |
| 4 | 4653.69 | 1814.64 (±150.1) | 1746.66 (±173.3) | **4061.33** (±421.1) |
| 5 | 5357.37 | 2091.17 (±168.6) | 2014.28 (±199.7) | **4287.24** (±434.4) |

As we previously explained, FA-OPD overcomes the limitation of lack-of-online-exploration inherent in traditional flow-matching and diffusion policies, which becomes catastrophic when expert demonstrations are sub-optimal. This section validates that claim by evaluating the robustness of FA-OPD, FP, and DP to sub-optimal expert data in the Walker2d environment. Specifically, we trained each algorithm using expert demonstrations with varying levels of episode return and compared the converged performance of the learned policies. As shown in Table 1, when expert returns are extremely low (ID: 1), DP slightly outperforms both FP and FA-OPD. This is expected and exposes a known boundary of any IRL method that learns its reward purely from expert data: with near-novice demonstrations (Expert Return ≈ 265), the FM discriminator cannot recover a meaningful density-ratio signal, and the resulting reward is essentially uninformative for online exploration. The diffusion / flow behavioral cloning baselines fare slightly better here only because they directly imitate the dataset, regardless of its quality. As soon as the expert exceeds this novice threshold (ID ≥ 2), the situation reverses sharply and FA-OPD takes a substantial lead. However, once the expert performance exceeds a novice level (i.e., achieves relatively higher returns), FA-OPD significantly surpasses both DP and FP (IDs: 2–5), and even achieves beyond-expert performance in some cases (IDs: 2–3). These results demonstrate the robustness of FA-OPD to sub-optimal expert data. In contrast, FP and DP, which lack online interaction, are prone to overfitting to the expert data. This is particularly detrimental when the data quality is poor. FA-OPD remains robust under the same conditions because the expert data only indirectly guides online exploration and policy updates through the reward signal: it influences the "teacher" FM model without directly constraining the agent's simple MLP policy.

## 4.4. [RQ 4.] Case Study: FM policy with online RL

Given the poor robustness of FM policies in offline settings with sub-optimal expert data, a natural question is whether we can instead update an FM policy online with policy gradients. While theoretically feasible, this introduces substantial instability and extra computational overhead (see Appendix C). We empirically validate this in Maze2d by comparing FA-OPD's training curves and wall-clock time (Figure 4) against two naive baselines we proposed and one existing work that all couple online RL with an FM policy: FM-A2C (baseline), which updates the FM policy via an actor-critic framework using reparameterization without explicit density evaluation; FM-PPO (baseline), which applies Proximal Policy Optimization with reparameterization and requires explicit density computation for importance sampling; and FPO (McAllister et al., 2025), which uses the FM loss to approximate a proxy importance ratio and then performs a standard PPO update.

As shown in Figure 4a, three alternative ways for online updating FM policies via policy gradient fail to learn a feasible policy. Among these, FM-PPO appears to be the most promising one, reaching a peak success rate of near 0.33, but rapidly declines due to instability caused by backpropagation through time (BPTT). In contrast, FA-OPD avoids the instability of FM policy's gradient computation by distilling a training-time FM teacher into a student policy. This policy can be updated stably using standard policy gradient methods, thereby achieving consistent and superior performance. Figure 4b compares the training time (for 8192 training epoches) and inference time (over 40k transition steps) of each algorithm. In terms of training cost, both FM-PPO and FPO are considerably more costly: FM-PPO requires Hutchinson trace estimation (Lipman et al., 2022) to approximate probability densities, while FPO must compute FM losses for both old and new policies to estimate importance sampling ratios for each sample—both of which are computationally intensive. In contrast, FA-OPD is significantly more efficient due to its architectural design that minimizes overhead. During inference, FM-PPO, FM-A2C and FPO have similarly high time cost since they use FM policies

with the same architecture. In contrast, FA-OPD has significantly lower time cost. The key reason is that the other three methods adopt FM policy that requires multi-step numerical integration for action generation, while FA-OPD's behavioral policy is a simple MLP-based policy which is as capable as the FM policy in terms of performance.

Notably, since FPO assumes the RL setting, namely we have access to the true reward of environment, we conduct the experiments of these three baselines (FPO, FM-A2C and FM-PPO) with access to true reward. Although FA-OPD follows the IRL setting without access to the true reward, the comparison is still meaningful because all four methods perform online policy improvement driven by a reward signal. For fair comparison, we conducted additional experiments with the unified learned reward in Appendix F.4.

## 5. Related Work

Diffusion policy (Ankile et al., 2024; Chi et al., 2024; Pearce et al., 2023; Reuss et al., 2023; Sridhar et al., 2023; Ze et al., 2024) pioneers the use of modern generative models for policy representation, advancing the ability to capture complex action distributions beyond Gaussian approximations. Following the success of diffusion models, Flow Matching (FM) offers a promising alternative that enables efficient training, fast sampling, and improved generalization compared to diffusion models (Lipman et al., 2023; Zheng et al., 2023), and has gained popularity in robot learning (Braun et al., 2024; Zhang & Gienger, 2025), image generation (Lipman et al., 2023; Esser et al., 2024), and video synthesis (Kong et al., 2025; Wan et al., 2025a).

However, traditional diffusion or FM policies are typically designed for offline settings to clone expert data through supervised learning. Recently, several works attempted to adapt diffusion policies to reinforcement learning beyond mere behavioral cloning. For example, DQL (Wang et al., 2023) and IDQL (Hansen-Estruch et al., 2023) employ diffusion models to represent the policy network and use an actor-critic architecture (Konda & Tsitsiklis, 1999) to perform policy-gradient optimization. Nevertheless, these methods suffer from extreme instability during policy gradient computation due to the long-chain structure of the diffusion policy. FQL (Park et al., 2025) follows a similar approach but uses an FM policy and transfers knowledge from a full FM model into a simpler one to improve stability. However, these methods are designed only for offline settings and lack support for exploration in online environments.

To adapt the strong distribution-matching capability of diffusion or FM policies to online settings, DIPO (Yang et al., 2023) introduces action gradients to directly update action

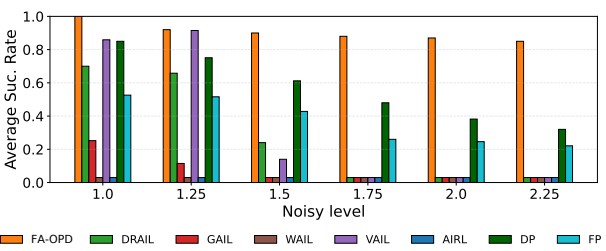

*Figure 3.* Performance of all methods in Fetch-pick environment across 6 noisy-levels.

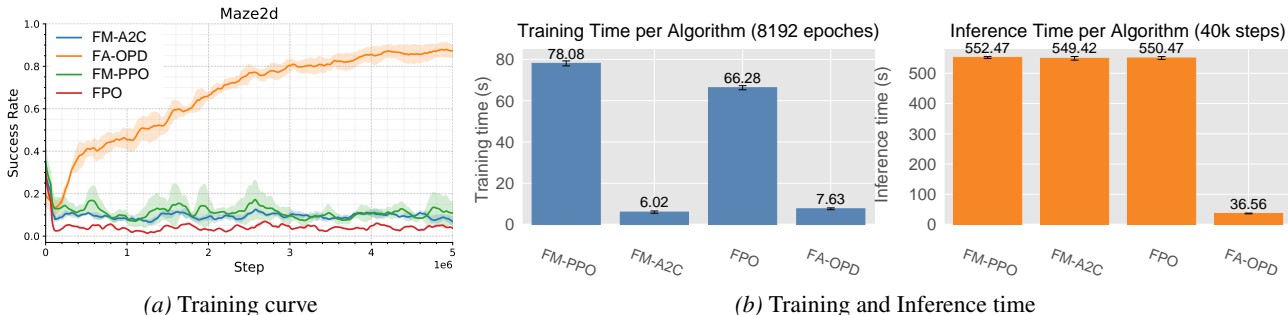

*(a)* Training curve        *(b)* Training and Inference time

*Figure 4.* Comparison of four algorithms for online updating FM policies. The left figure (a) shows the training curve of four algorithms in Maze2d, and the right figure (b) compares the computational overhead (time cost) of four algorithms.

batches based on the Q-function and re-fits the distribution of new action batches, but is limited to purely value-based reinforcement learning. Alternatively, QSM (Psenka et al., 2025) and DPPO (Ren et al., 2024) use policy gradients to fine-tune pre-trained diffusion policies in online environments. Similar to DPPO, ReinFlow (Zhang et al., 2025b) pre-trains a flow-matching policy and fine-tunes it in online environments. Recent work further studies latent-space optimization or spatially aware RL adaptation for diffusion/FM policies (Zhang et al., 2026; Pan et al., 2026). Flow-GRPO (Liu et al., 2025) and ORW-CFM-W2 (Fan et al., 2025) also use RL to fine-tune flow matching models, but with a focus on vision-based tasks rather than policy modeling. Our method eliminates the need for post-training fine-tuning and enables learning a policy from scratch that benefits from the distribution-matching capability and efficiency of Flow Matching Model.

Similar to ours, many IRL methods learn a reward to guide online exploration. The most established is the GAIL family (Ho & Ermon, 2016)—including GAIL-GP, VAIL, and WAIL (Peng et al., 2018; Xiao et al., 2019)—which pioneered adversarial imitation learning; related AIL formulations have also been used for sparse and noisy sequential data (Wan et al., 2025b). However, GAIL-style methods typically use simple discriminators that do not explicitly model distributions, often yielding imprecise rewards for behaviors close to expert demonstrations. To address this, DiffAIL (Wang et al., 2024) and DRAIL (Lai et al., 2024) adopt diffusion-based discriminators to improve distribution matching capabilities. Despite better discrimination, they are computationally inefficient and tend to poorly calibrate rewards on out-of-distribution (OOD) state-action pairs. FA-OPD overcomes these issues by using an optimal-transport Flow Matching discriminator that is significantly more efficient and robust.

**On-policy distillation.** A related line of work uses on-policy distillation to train a student model on its own rollouts under supervision from a teacher (Agarwal et al., 2023; 2024). These methods have been behind much of the recent

progress in LLM post-training, and they assume a strong, fixed pre-trained teacher (a larger LLM, a value model, or a fitted reward model). The OPD literature splits into two supervision modes: reward-based, where the teacher's score becomes a per-action RL reward, and target-based, where the teacher's output becomes a direct regression target. DAgger (Ross et al., 2011) is the canonical example of the target-based mode in control. FA-OPD combines *both* modes under a single co-trained teacher (Sec. 3.1). The teacher is learned from demonstrations rather than pre-trained, which our experiments show is important: a frozen pre-trained teacher gives uncalibrated scores on the student's changing on-policy distribution, while a co-trained teacher stays sharp at the expert/student boundary throughout learning.

Table 7 in Appendix H summarizes the properties of existing works.

## 6. Conclusion

In this work, we identify the key limitations of behavior-cloning-based FM policies, and the key challenges of applying FM policies in online reinforcement learning. Then, we propose a novel framework, Flow Adversarial On-Policy Distillation (FA-OPD), that enables effective online policy updates using expert demonstrations while enabling the policy to benefit from the strength of FM. Extensive experiments across robot navigation, manipulation, and locomotion tasks validate the effectiveness, generalizability, robustness, and efficiency of the proposed approach. The source code of this project is provided at https://github.com/vanzll/FA-OPD.

## Acknowledgements

This research/project is supported by the National Research Foundation, Singapore under its National Large Language Models Funding Initiative (AISG Award No: AISG-NMLP-2024-003). Any opinions, findings and conclusions or recommendations expressed in this material are those of the author(s) and do not reflect the views of National Research

Foundation, Singapore.

## Impact Statement

This paper introduces Flow Adversarial On-Policy Distillation (FA-OPD), a framework designed to improve efficiency, robustness, and generalization in learning from expert demonstrations. By addressing limitations in existing adversarial imitation learning methods, FA-OPD enables more stable reward guidance and mitigates risks of misestimation in out-of-distribution states.

The potential broader impacts of this work include advancing safe and reliable reinforcement learning for robotics, autonomous systems, and AI agents. Applications such as robot navigation, manipulation, and autonomous driving could benefit from improved learning efficiency and robustness, contributing to safer deployment in real-world environments. Ethical considerations include ensuring that such systems are used responsibly, with attention to safety, fairness, and transparency.

Overall, this paper aims to advance the field of Machine Learning. While many societal consequences are possible, we believe none require specific emphasis beyond the general importance of developing trustworthy and efficient learning frameworks.

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

# A. Discussion

This sections provides the answers to possible questions about FA-OPD framework and the experiment results in Q&A format below.

## A.1. Query about Experiment Results

Q&A

**Q. Why the performance improvement of FA-OPD is marginal in Navigation tasks?**

**A.** This observation exactly aligns with the motivation of FA-OPD. Please note that the advantage of our FM-enhanced discriminator lies in capturing multi-modal expert distributions during training, and our action-distillation term aims to stabilize the training process. In navigation tasks, the objective is to move the agent towards a target position. We explained in the experiment session that the success trajectory is often unique (uni-modal) and is fully covered by expert data, obviating the need of excessive exploration and complex, multi-modal training. Therefore, FA-OPD shows marginal improvement but the experiments also proves in such relatively "uni-modal tasks", FA-OPD roughly degraded to other baselines but with greater stability (reflected by lower training variance), preserving its lower-bound performance. In Manipulation tasks like Hand-rotate and Fetch-pick, the goal is not anymore "moving to a target place" but to try to manipulate an object, enabling more complexity, flexibility and multi-modality in expert data. In such tasks, FA-OPD significantly outperforms other methods, validating our motivation.

**Q. The experiment result of robustness study (section 4.3) does not support the claim that FA-OPD overcomes the limitation of suboptimal expert data, as they are approximately or often below the expected return of demonstration data.**

**A.** Actually, the claim of "robustness to sub-optimal data" is relative to baseline, rather than expert performance. Since our method is inherently imitation learning, the policy is optimized absolutely from expert data. Therefore, the return of learned policy can hardly surpass expert demonstration. However, when experts are sub-optimal, our method exhibits much stronger robustness compared to DP and FP, highlighting the importance of online exploration which forms the motivation of our method.

## A.2. Query about FA-OPD framework

Q&A

**Q. Why do you use a complex FM model as the "teacher" but only use a simple MLP-based policy as the "student"?**

**A.** This design leverages the advantages of the FM model in policy representation while addressing the limitations of directly applying FM to policy modeling. Specifically, the FM policy is hard to update using online RL algorithms due to its architecture, whereas our "student" MLP policy can explore the environment and be updated online efficiently thanks to its simple structure. Furthermore, by distilling the knowledge of the "teacher" FM model into the "student" MLP policy, the agent benefits from both fast inference and an awareness of the expert data distribution.

**Q. Why do you model the joint distribution of $s$ and $a$ in the "teacher" FM model, rather than modeling the conditional distribution $a \mid s$ as in the traditional FM policy paradigm?**

**A.** First, the "teacher" FM model is used only during training, so it need not be a deployed conditional policy. Second, the reward-distillation channel must judge complete state–action pairs, not actions alone: an action can be expert-like in one state and poor in another. Modeling the joint distribution therefore gives the FM scorer access to the occupancy structure that adversarial imitation learning tries to match. For action distillation, we query the same FM teacher at student-visited states to obtain expert-like target actions, so the student receives local corrections while the reward channel still evaluates whether the resulting state–action visits remain close to the expert distribution.

**Q. How should we understand "distillation" in FA-OPD?**

**A.** The two distillation modes of FA-OPD transfer teacher knowledge into the student in complementary ways. (1) *Reward distillation* is **indirect**: the FM teacher's per-action score is converted into an RL reward, and the student is updated to maximize cumulative reward via PPO. (2) *Action distillation* is **direct**: the FM teacher generates a target action at each student-visited state,

and the student is regressed onto it via MSE. Together they teach the student through both reward-guided online improvement and dense local action targets.

**Q. Is your "action distillation" the same as DAgger?**

**A.** Spiritually yes. Both supervise the student at the states the student itself visits, using teacher-supplied actions as labels, which is the standard fix for the covariate-shift problem of offline behavior cloning (Ross et al., 2011). The key technical difference is that classical DAgger requires an *oracle expert* that can be queried at arbitrary states, which is not available in our setting. FA-OPD's teacher is a flow-matching generator that has *learned* to produce target actions from a static demonstration dataset, and is co-trained with the student. Action distillation in FA-OPD thus inherits DAgger's covariate-shift resistance without needing an interactive expert.

**Q. If the deployed student is a unimodal Gaussian, why does a multi-modal teacher matter?**

**A.** *Multimodal training ≠ multimodal deployment.* Consider expert demonstrations with two distinct strategies through a maze, left-first (40%) and right-first (60%). A unimodal AIL discriminator (e.g., GAIL's MLP) cannot resolve which mode a given $(s, a)$ belongs to, yielding a coarse reward and **mode-averaging failures** (straight-first). FA-OPD's FM teacher instead learns the full multi-modal expert distribution, and its per-action score recognizes which mode the current behavior resembles and how closely. The student then uses this multimodal-informed reward to discover and commit to **one optimal mode** through online exploration, rather than averaging across incompatible modes. This mirrors classical knowledge distillation (Hinton et al., 2015): a complex teacher's soft outputs carry rich inter-class structure that lets a simpler student match its quality.

**Q. Does online interaction truly provide benefits when rewards come solely from offline data? Since the reward function is learned from an offline dataset, it can only encourage in-distribution behavior, and action distillation further restricts exploration, making the framework essentially mimic the offline dataset through joint optimization.**

**A.** The reward (computed from the FM teacher's score) trains on offline data, but it learns a **generalizable function** that evaluates arbitrary $(s, a)$ pairs, including unseen ones. Consider RLHF: it learns from a fixed human feedback dataset yet lets models generalize to novel prompts (Ouyang et al., 2022). The teacher's score is a learned abstraction of what makes behavior "expert-like," not a memorization of specific $(s, a)$ pairs. Saying we "mimic the offline dataset" oversimplifies what is happening. We are learning the *expert's task objective*, which is different from BC. Behavioral cloning methods (Diffusion Policy, Flow Matching Policy) directly memorize $(s, a)$ mappings and generalize poorly. GAIL matches state-action occupancy measures point-wise, which also limits generalization. FA-OPD learns the *distributional structure* of expert data, which lets the policy understand what makes expert behavior successful and whether new $(s, a)$ pairs match this objective. In some experiments our learned policies even exceed expert performance, suggesting we have captured something deeper than surface behavior, likely because expert demonstrations contain suboptimal actions due to human error or limited data. FM captures the "general strategy" behind expertise. In this view, reward distillation is the exploration-driving component: it gives PPO a learned objective that can generalize beyond point-wise demonstrations. Action distillation is the exploitation-stabilizing component: it gives dense local targets at student-visited states and prevents reward-guided exploration from drifting into regions where the learned reward becomes uninformative.

# B. Pseudo Code

This section shows the implementation Pseudo Code of FA-OPD algorithm.

---

**Algorithm 1 Flow Adversarial On-Policy Distillation (FA-OPD)**

---

1: **Input:** Expert trajectory dataset $D_E$, initial agent policy $\pi_\theta$, initial FM generator $G_\theta$
2: **Initialize:** Student policy parameters $\phi$, FM generator parameters $\theta$
3: **while** not converged **do**
4:     Sample expert trajectories $\tau_E \sim D_E$
5:     Roll out agent policy $\pi_\theta$ to obtain trajectories $\tau_A$

6:     **Student Policy Update:**
7:         Compute reward $\nabla_\theta(s, a)$ for $(s, a)$ in $\tau_A$ via Eq. 6
8:         Compute policy loss $\mathcal{J}(\phi)$ based on policy optimization objective (Eq. 11)
9:         Update policy parameters:
10:             $\phi \leftarrow \phi + \alpha \nabla_\phi \mathcal{J}(\phi)$

11:     **Teacher FM Model Update:**
12:         Compute FM discriminator loss $\mathcal{L}_{\mathrm{FM}}(\theta)$ based on Eq. 5, 9 and 10
13:         Update FM generator parameters:
14:             $\theta \leftarrow \theta - \gamma \nabla_\theta \mathcal{L}_{\mathrm{FM}}(\theta)$
15: **end while**
16: **Output:** Optimized agent policy $\pi_{\phi^*}$ and FM generator $G_{\theta^*}$

---

## C. Why an FM policy is hard to update in the traditional policy–gradient paradigm

In this section, we will provide mathematical and statistical insight for readers to understand why FM policy is hard to update in online reinforcement learning via the traditional policy-gradient paradigm.

**Setup.** Let $\pi_\theta(a \mid s)$ be a stochastic policy. We define the objective as

$$J(\theta) = \mathbb{E}_{(s,a)\sim\pi_\theta}\big[Q(s,a)\big], \tag{12}$$

where $Q(s, a)$ denotes a state–action value (critic). In the policy–gradient framework we are interested in its gradient

$$\nabla_\theta J(\theta) = \nabla_\theta \mathbb{E}_{(s,a)\sim\pi_\theta}\big[Q(s,a)\big]. \tag{13}$$

Since $\theta$ appears inside the sampling distribution $\pi_\theta$, this gradient is not straightforward to compute. In the traditional case, three standard approaches are commonly used: (1) likelihood–ratio, (2) explicit reparameterization (pathwise), and (3) implicit reparameterization (via the CDF). We briefly recall them, then explain why each becomes difficult for a Flow–Matching (FM) policy.

**Route I: Likelihood–ratio (LR).**
$$\nabla_\theta J(\theta) = \mathbb{E}[\nabla_\theta \log \pi_\theta(a \mid s)\, Q(s,a)]. \tag{14}$$

For a Gaussian with mean $\mu_\theta(s)$ and covariance $\Sigma_\theta(s)$, the log-likelihood can be written in closed form as

$$\log \pi_\theta(a \mid s) = -\frac{1}{2}\Big((a - \mu_\theta)^\top \Sigma_\theta^{-1}(a - \mu_\theta) + \mathrm{logdet}(2\pi\Sigma_\theta)\Big), \tag{15}$$

so both $\log \pi_\theta$ and $\nabla_\theta \log \pi_\theta$ are easy to compute and numerically stable.

**Route II: Explicit reparameterization (pathwise).** Introduce a standard Gaussian noise $\xi \sim \mathcal{N}(0, I)$ with the same dimension as the action. The action can then be written as

$$a = g_\theta(s, \xi) = \mu_\theta(s) + L_\theta(s)\,\xi, \qquad L_\theta(s)\,L_\theta(s)^\top = \Sigma_\theta(s), \tag{16}$$

With this reparameterization, we have

$$\nabla_\theta J(\theta) = \mathbb{E}_{s,\xi}\left[\nabla_a Q(s,a)\, \frac{\partial g_\theta(s,\xi)}{\partial \theta}\right]. \tag{17}$$

Since $g_\theta$ is affine in the Gaussian case (a linear transform plus a shift), the backpropagation path involves only a single matrix–vector multiplication and addition. As a result, the gradient is inexpensive to compute and numerically stable.

**Route III: Implicit reparameterization gradients (IRG).**   When a distribution does not have a simple reparameterization map $g_\theta(s, \xi)$, one can instead differentiate through its cumulative distribution function (CDF) $F_\theta$. The idea is to use the inverse-CDF sampling scheme: draw $u \sim \mathrm{Unif}[0, 1]$ and define the sample $a$ implicitly by

$$F_\theta(a) = u.$$

Since $u$ is fixed once drawn, differentiating this relation with respect to $\theta$ (via the implicit function theorem) yields, in one dimension,

$$\frac{\partial a}{\partial \theta} = -\left( \frac{\partial F_\theta}{\partial a} \right)^{-1} \frac{\partial F_\theta}{\partial \theta}. \tag{18}$$

Here $\frac{\partial F_\theta}{\partial a}$ is exactly the density $p_\theta(a)$ evaluated at the sample, while $\frac{\partial F_\theta}{\partial \theta}$ captures how the CDF changes as the parameters $\theta$ vary. Intuitively, as $\theta$ changes, the sampled point $a$ must shift in order to preserve the same CDF value $u$.

In higher dimensions, a single CDF is not enough; instead one introduces a sequence of conditional CDFs, known as the *Rosenblatt transform*, to map a vector of uniform random variables into a valid sample. This provides a general way to obtain reparameterization gradients even for distributions that lack an explicit sampler. In practice, however, for distributions like the Gaussian, this route is rarely used because an explicit affine reparameterization is already available (Route II).

**FM policies in a line.**   An FM policy samples actions by integrating a conditional ODE flow:

$$\frac{dx_t}{dt} = v_\theta(x_t, s, t), \quad x_0 = \xi \sim p_0, \quad a = x_1. \tag{19}$$

Let $a = \Phi_{0 \to 1}^{\theta, s}(\xi)$ so that $\pi_\theta(\cdot \mid s) = (\Phi_{0 \to 1}^{\theta, s})_\# p_0$ (the pushforward of $p_0$ by the flow map). Sometimes the flow is trained with an explicit density (a CNF), sometimes by velocity regression without a likelihood ("pure FM").

Why the three traditional routes become difficult for FM policy

**LR for FM: likelihoods are no longer easy to compute.**   For CNFs, the log–density along a trajectory satisfies

$$\frac{d}{dt} \log p_t(x_t) = -\mathrm{tr}\left( \frac{\partial v_\theta}{\partial x}(x_t, s, t) \right), \tag{20}$$

$$\implies \quad \log \pi_\theta(a \mid s) = \log p_0(\xi) - \int_0^1 \mathrm{tr}\left( \frac{\partial v_\theta}{\partial x}(x_t, s, t) \right) dt, \qquad a = \Phi_{0 \to 1}^{\theta, s}(\xi). \tag{21}$$

Thus, to obtain $\log \pi_\theta(a \mid s)$ for one sample we must (i) solve the ODE to obtain the path $x_{0:1}$, (ii) compute a trace integral along that path (often with a Hutchinson estimator, which needs multiple Jacobian–vector or vector–Jacobian products), and then (iii) differentiate this pipeline with respect to $\theta$. Since LR needs $\nabla_\theta \log \pi_\theta$, it inherits the same pipeline. Let $N_{\mathrm{fe}}$ be the average number of vector–field evaluations per ODE solve, $K$ the number of probe vectors for the trace, $B$ the batch size, and $H$ the rollout horizon. A single on–policy update already needs roughly

$$\Omega\big( B \, H \, N_{\mathrm{fe}} \, (1 + K) \big) \tag{22}$$

forward evaluations, before parameter backpropagation. In "pure FM", $\log \pi_\theta$ is not available at all, so LR is inapplicable.

**IRG for FM: the needed CDFs are not exposed.**   Implicit reparameterization requires access to $F_\theta$ (and, in multiple dimensions, conditional CDFs). An FM policy provides an implicit sampler $a = \Phi_{0 \to 1}^{\theta, s}(\xi)$, not a tractable $F_\theta$. Forcing a CNF only to recover $F_\theta$ brings us back to the same ODE solves and trace integrals as LR. The obstacle is structural.

**Pathwise for FM: reasonable in form, but it brings BPTT.**   Formally, FM policies are already reparameterized. Given noise $\xi \sim p_0$ and flow map $\Phi_{0 \to 1}^{\theta, s}$, the action is $a = \Phi_{0 \to 1}^{\theta, s}(\xi)$. The objective and its gradient can be written as

$$J(\theta) = \mathbb{E}_{s, \xi}\left[ Q\left( s, \Phi_{0 \to 1}^{\theta, s}(\xi) \right) \right], \tag{23}$$

$$\nabla_\theta J(\theta) = \mathbb{E}_{s, \xi}\left[ \nabla_a Q(s, a) \, \frac{\partial \Phi_{0 \to 1}^{\theta, s}(\xi)}{\partial \theta} \right], \qquad a = \Phi_{0 \to 1}^{\theta, s}(\xi). \tag{24}$$

Thus the central difficulty is computing the sensitivity $\partial\Phi/\partial\theta$, i.e. how the final action changes when the policy parameters change. There are two standard approaches:

*(i) Backpropagation Through Time (BPTT).* Here one treats the numerical ODE solver as a sequence of discrete updates

$$x_{k+1} = \Phi_k(x_k;\theta), \qquad k = 0,\dots,N-1, \qquad a = x_N,$$

and attaches a terminal loss $L = \ell(a)$. Gradients are then propagated backward step by step, like training a recurrent neural network:

$$\lambda_N = \nabla_{x_N}\ell, \qquad \lambda_k = \left(\tfrac{\partial\Phi_k}{\partial x_k}\right)^\top \lambda_{k+1}.$$

The overall parameter gradient accumulates as

$$\nabla_\theta L = \sum_{k=0}^{N-1} \left(\tfrac{\partial\Phi_k}{\partial\theta}\right)^\top \lambda_{k+1}.$$

*(ii) Continuous adjoints / sensitivities.* Instead of unrolling the solver, one can differentiate the continuous ODE system directly. In the forward sensitivity method, one integrates the matrix

$$S_t = \frac{\partial x_t}{\partial\theta}, \quad \frac{dS_t}{dt} = \frac{\partial v_\theta}{\partial x}(x_t,s,t)\,S_t + \frac{\partial v_\theta}{\partial\theta}(x_t,s,t), \quad S_0 = 0,$$

and obtains $\partial a/\partial\theta = S_1$ at the end of the flow. Alternatively, the adjoint method integrates a backward variable $\lambda_t$, yielding

$$\nabla_\theta J(\theta) = \mathbb{E}\left[\int_0^1 \left(\tfrac{\partial v_\theta}{\partial\theta}(x_t,s,t)\right)^\top \lambda_t\,dt\right].$$

Both formulations are mathematically equivalent; the choice depends on whether one prefers forward or backward accumulation of sensitivities.

**Why this still struggles in practice.** Compared to the Gaussian case, computing sensitivities for FM policies is significantly more involved because the gradient path runs through an entire ODE solver rather than a short affine transformation.

*Cost.* Each sample requires both a forward ODE solve and a backward pass of similar or higher cost. With adaptive solvers or mildly stiff dynamics, the average number of function evaluations $N_{\mathrm{fe}}$ grows, making training expensive. The complexity of one update scales roughly as

$$\widetilde{O}\big(B\,H\,N_{\mathrm{fe}}\,C_{\mathrm{jvp/vjp}}\big),$$

where $B$ is the batch size, $H$ is the rollout horizon, and $C_{\mathrm{jvp/vjp}}$ is the cost of a Jacobian–vector or vector–Jacobian product. Continuous adjoints save memory but not time; in stiff regimes, the backward solve can even be harder than the forward one.

*Numerical mismatch.* Continuous adjoints differentiate the *continuous* ODE, while the forward pass uses a *discretized* solver. With adaptive step sizes or differing time grids, the two gradients can diverge, hurting stability. Using discrete adjoints (matching the backward steps to the forward solver) or checkpointing can reduce this mismatch, but both add implementation effort and runtime overhead.

*Variance.* The gradient estimator combines the critic's action gradient $\nabla_a Q(s,a)$ with the flow sensitivity $\partial a/\partial\theta$. For actions generated by a deep ODE chain, the sensitivity can be large and noisy, amplifying variance and making optimization unstable. In practice this often necessitates smaller learning rates, stronger target networks, and additional regularization, which lowers sample efficiency.

**Conclusion.** In Gaussian policies, likelihoods are easy to compute, the reparameterization path is short, and CDFs are tractable, so all three routes are practical. In FM policies, however, LR requires ODE solves and trace integrals, IRG depends on CDFs that are not exposed, and the pathwise route—though natural in form—forces gradients through an ODE solver, leading to high cost, numerical issues, and higher variance. The gradients do exist, but in high-dimensional, long-horizon, online settings they are rarely a favorable trade-off in practice.

# D. Theoretical Justification of the FM-Enhanced Discriminator

This appendix formalizes why the FM-enhanced discriminator (Eq. 10) yields a principled reward signal for AIRL, rather than a heuristic distance, thus providing theoretical understanding of why a flow-based discriminator outperforms its diffusion-based counterpart **given the same computational budget**. The argument proceeds in three steps: (i) the per-sample FM loss is, up to a data-only constant, the negative of a variational lower bound on the model log-likelihood (Proposition D.1); (ii) the AIRL log-ratio reward built from FM losses asymptotically recovers the log density ratio between the expert and agent conditional models (Proposition D.2); and (iii) under OT-conditional paths, the resulting bound tends to be tighter and lower-variance than the score-based bound used by diffusion-based discriminators (Remark D.3).

We adopt the conditional FM model and notation of Sec. E.1.1, and recall the per-sample FM distance $Dist_\theta(x \mid c)$ from Eq. (9).

**Proposition D.1** (FM loss as a negative ELBO). *For any point $x$ and condition $c$, there exists a constant $C(x)$ independent of $\theta$ such that*

$$-\tfrac{1}{2} Dist_\theta(x \mid c) \; = \; \mathrm{ELBO}_\theta(x \mid c) \; - \; C(x), \qquad \mathrm{ELBO}_\theta(x \mid c) \; \leq \; \log p_\theta(x \mid c). \qquad (25)$$

This follows by specializing the weighted-MSE–ELBO equivalence of Kingma & Gao (2023) to the constant-variance OT path of Lipman et al. (2023); McAllister et al. (2025, Prop. 1) state the same equivalence directly in the flow-matching regime. Operationally, smaller Dist implies a tighter likelihood lower bound, so Dist is a calibrated proxy for the negative log-likelihood under the $c$-conditioned model.

**Proposition D.2** (AIRL reward as a log density ratio). *Let $D_{FM,\theta}$ be the Softmax discriminator of Eq. (10). The AIRL log-ratio reward (Eq. (6)) simplifies algebraically to*

$$r_\theta(s, a) \; = \; \frac{Dist_\theta(s, a \mid c{=}0) - Dist_\theta(s, a \mid c{=}1)}{\tau}, \qquad (26)$$

*and converges, as the conditional FM models fit $\rho_E$ and $\rho_\pi$, to the maximum-entropy IRL reward of Fu et al. (2017):*

$$\tfrac{\tau}{2} r_\theta(s, a) \; \longrightarrow \; \log \rho_E(s, a)/\rho_\pi(s, a). \qquad (27)$$

The algebraic step is immediate (the Softmax partition cancels in Eq. (6)); the asymptotic limit follows by combining Proposition D.1 with the standard density-ratio argument of Fu et al. (2017, Prop. 1). The temperature $\tau$ only rescales the reward, which is absorbed by PPO as a reward-scale choice in practice. In other words, FA-OPD inherits AIRL's reward functional and merely re-parameterizes it through a density-aware quantity, which helps the reward remain calibrated on OOD pairs that are far from both modeled distributions.

*Remark* D.3 (OT paths shrink the ELBO gap). The gap $\log p_\theta(x) - \mathrm{ELBO}_\theta(x)$ is controlled by the KL between the approximate and true reverse posteriors along the chosen probability path. OT-conditional paths minimize this transport cost (Lipman et al., 2023), and the resulting time-constant target velocity $u_t = x_1 - x_0$ removes the time-dependent score-scaling that amplifies variance in DSM-based losses (McAllister et al., 2025, discussed in). Together, these properties make $Dist_\theta$ a tighter and lower-variance proxy for distributional mismatch than the diffusion-based losses used by DiffAIL (Wang et al., 2024) and DRAIL (Lai et al., 2024), consistent with the empirical gap in Table 3. Under the OPD lens of Sec. 2.3, Propositions D.1–D.2 together justify the FM teacher as a *well-calibrated OPD scorer*: its per-action score is tightly tied to expert log-likelihood, and the bound is tighter than that of any diffusion-based teacher under the same compute budget.

# E. Implementation Details

The experiment environments are customized and adapted from widely used platforms, including OpenAI Gymnasium, (Towers et al., 2024), D4RL (Fu et al., 2021), and MuJoCo (Todorov et al., 2012). Part of the baseline RL, IL, and IRL implementations are adapted from rl-toolkit (Szot, 2024). The implementation of noisy environment in section 7 are adapted from Goal-prox-il (Lee et al., 2021) and DRAIL (Lai et al., 2024).

All experiments are conducted on a Linux server equipped with four NVIDIA A40 (48GB) GPUs and an AMD EPYC 7543P 32-core CPU.

We show the algorithmic and experimental implementation details below.

### E.1. Algorithmic Details

#### E.1.1. CHOICE OF CONDITIONAL PROBABILITY PATHS IN FM

We model the joint vector $x = (s, a)$ and condition the model on $c \in \{0, 1\}$ (e.g., $c{=}1$ for expert data, $c{=}0$ for agent data). For the conditional probability path, we use a simple straight-line (i.e. Optimal Transport path (Lipman et al., 2022)) conditional path between a noise start point and a target joint sample:

$$x_0 \sim \mathcal{N}(0, I), \qquad x_t = (1 - t)\, x_0 + t\, x_1, \quad t \in [0, 1], \quad x_1 = (s_1, a_1). \tag{28}$$

Under this path, the target velocity is time-constant:

$$u_t(x_t \mid x_1, c) = x_1 - x_0, \tag{29}$$

while the model predicts $v_\theta(x_t, t \mid c)$. The corresponding Flow Matching loss is:

$$\mathcal{L}_{\text{FM}}(\theta) = \mathbb{E}_{\substack{(x_1,c)\sim\mathcal{D},\, x_0\sim\mathcal{N}(0,I) \\ t\sim\mathcal{U}[0,1]}} \left\| v_\theta(x_t, t \mid c) - (x_1 - x_0) \right\|^2, \qquad x_t = (1 - t)x_0 + tx_1. \tag{30}$$

Optimal Transport paths offer key advantages in probability path construction. First, OT paths form the shortest geodesic connections between distributions, mathematically achieved by minimizing the transport cost. This shortest-path property yields two central benefits: it significantly improves training efficiency, as straighter trajectories result in lower variance and faster convergence; and it enhances sample quality by better preserving structural features of the target distribution.

#### E.1.2. COMPUTATION OF DIST IN THE FM-ENHANCED DISCRIMINATOR

We use the FM model over the joint input $x = (s, a)$ and condition on $c \in \{0, 1\}$ (expert: $c{=}1$, agent: $c{=}0$). Given a target pair $x_1 = (s, a)$, we sample a noise start $x_0 \sim \mathcal{N}(0, I)$ and define the straight-line conditional path

$$x_t = (1 - t)\, x_0 + t\, x_1, \qquad t \in [0, 1], \tag{31}$$

with target velocity

$$u_t(x_t \mid x_1, c) = x_1 - x_0. \tag{32}$$

The FM distance (our "loss") at $(s, a)$ under condition $c$ is the expectation of the per-sample discrepancy:

$$\text{Dist}_\theta(s, a \mid c) = \mathbb{E}_{t\sim\mathcal{U}[0,1],\, x_0\sim\mathcal{N}(0,I)} \left[ \left\| v_\theta(x_t, t \mid c) - (x_1 - x_0) \right\|_2^2 \right], \quad x_t = (1 - t)x_0 + tx_1. \tag{33}$$

**Monte Carlo estimation.** We approximate the expectation by Monte Carlo. With $S$ samples $\{(t_i, x_0^{(i)})\}_{i=1}^S$,

$$\widehat{\text{Dist}}_\theta(s, a \mid c) = \frac{1}{S} \sum_{i=1}^S \left\| v_\theta(x_t^{(i)}, t_i \mid c) - (x_1 - x_0^{(i)}) \right\|_2^2, \qquad x_t^{(i)} = (1 - t_i)\, x_0^{(i)} + t_i\, x_1. \tag{34}$$

In practice we use stratified time sampling $t_i \in [0, 1]$ with small jitter and vectorize all $S$ samples across the batch for efficiency. For the discriminator branch during training, we also support the $S{=}1$ single-sample variant (one $t$ and one $x_0$ per $(s, a)$) for speed, which is an unbiased estimator of the expectation.

**From Dist to reward.** Given the label-conditioned distances, we form the FM-enhanced discriminator via a temperature-scaled Softmax over negative distances (consistent with Eq. 10):

$$D_{\text{FM},\theta}(s, a) = \frac{\exp\!\big(-\widehat{\text{Dist}}_\theta(s, a \mid c{=}1)/\tau\big)}{\exp\!\big(-\widehat{\text{Dist}}_\theta(s, a \mid c{=}1)/\tau\big) + \exp\!\big(-\widehat{\text{Dist}}_\theta(s, a \mid c{=}0)/\tau\big)}, \tag{35}$$

and compute the reward with the standard AIL/AIRL transform

$$r_\theta(s, a) = \log D_{\text{FM},\theta}(s, a) - \log\big(1 - D_{\text{FM},\theta}(s, a)\big). \tag{36}$$

Thus, smaller FM loss implies smaller Dist, larger $D_{\text{FM}}$, and a higher reward.

**Role of the temperature $\tau$ and typical Dist ranges.** Because $\text{Dist}_\theta(s, a \mid c)$ is a (scaled) negative ELBO (Appendix D, Proposition D.1), its scale is determined by the joint dimensionality $\dim(s) + \dim(a)$ and the noise scale used in the OT path, and is therefore environment-dependent. Empirically in our six benchmarks, expert points yield $\text{Dist} \in [10^{-1}, 1]$ once the FM model has converged, while early-training agent rollouts produce $\text{Dist} \gtrsim 1$ and occasionally an order of magnitude larger. A unit-temperature Softmax over such raw distances saturates to a hard $\{0, 1\}$ classifier on most $(s, a)$ pairs, which collapses the AIRL log-ratio reward to a flat $\pm\infty$ signal and destabilizes PPO updates. Introducing $\tau$ in Eq. 10 keeps the operating point of the Softmax in its informative regime regardless of the absolute distance scale, which we found removed the need for per-task tuning: a single $\tau = 0.1$ generalizes across all six environments. The value is reported in Table 2.

### E.1.3. IMPLEMENTATION PHILOSOPHY OF FA-OPD

A notable strength of FA-OPD is its algorithmic simplicity: the FM teacher changes the discriminator internals, but the deployed actor remains a standard Gaussian MLP optimized by PPO. Practitioners only need to implement $D_{\text{FM},\theta}$ while keeping interface compatibility with GAIL's discriminator API. The rest of the training pipeline (policy optimization, replay buffer management, and adversarial updates) is the same as vanilla GAIL.

When the action-distillation weight $\beta = 0$, the student strictly follows the standard PPO paradigm: a value/advantage function on the learned reward and a Gaussian MLP actor trained with the clipped PPO objective. When $\beta \neq 0$, we only add a single MSE term against teacher-generated targets, so the student can still be built on stock PPO with a modified reward and an action-distillation module.

These strengths collectively make FA-OPD accessible and easy to implement.

### E.1.4. POLICY ARCHITECTURE DETAILS

For all methods that use MLP policy architecture, we adopts a diagonal Gaussian policy: the policy is a diagonal Gaussian with a state-dependent learnable mean $\mu$ and single learnable log-std vector $\Sigma$ of size *action_dim*. $\Sigma$ is a global **state-independent** parameter, initialized to zeros (so std=exp(0)=1), broadcast across the batch, and optimized jointly with the actor via the PPO objective; it receives gradients from both the policy log-likelihood term and the entropy bonus in standard PPO paradigm. It is not a fixed constant and not state-dependent. Using a single global learnable log-std vector for PPO's diagonal Gaussian policy improves stability and simplicity: it decouples exploration scale from state, reducing gradient variance, typically yielding more stable, reproducible training and smoother convergence under PPO's clipped objective compared to state-dependent variance heads.

## E.2. Experimental Details

To ensure fair comparison, we adopt a practical two-tier strategy for hyperparameter configuration. For shared components, like policy and critic networks, learning rates, batch sizes, and discount factors, all methods use **identical settings**. This ensures that performance differences reflect genuine algorithmic improvements rather than implementation advantages. For FA-OPD's unique components (FM-enhanced discriminator), we deliberately choose **one straightforward set of hyperparameters and fix them across all tasks** without per-task tuning. This prioritizes demonstrating robustness and generality of the experiment results.

### E.2.1. HYPERPARAMETERS DETAILS OF FA-OPD

We summarize the hyperparameters used by FA-OPD in Table 2. They cover the FM-enhanced discriminator, the FM vector field, distance-based reward, and training logistics.

*Table 2.* Details of hyperparameters in FA-OPD.

| Name | Value | Meaning |
|---|---|---|
| fm_num_steps | 100 | FM time discretization steps (used for $t$ indexing in discriminator and for FM-based generation). |
| discrim_depth | 4 | Number of hidden layers in the FM discriminator's vector field $v_\theta$. |
| discrim_num_unit | 128 | Hidden width (units per layer) in $v_\theta$. |
| disc_lr | 1e-4 | Learning rate for the FM discriminator. |
| expert_loss_rate | 1.0 | Weight on expert branch loss term in discriminator training. |
| agent_loss_rate | -1.0 | Weight on agent branch loss term (negative encourages separation). |
| student_lr | 0.0001 | Learning rate of student policy. |
| reward_update_freq | 1 | Frequency to refresh rewards in the agent update loop (in updates). |
| state_norm | True | Whether to normalize states for the discriminator. |
| action_norm | False | Whether to normalize actions for the discriminator. |
| reward_norm | False | Whether to normalize rewards. |
| num_samples (MC) | 100 | Monte Carlo samples $S$ to estimate $Dist$ (expectation over $t$ and $x_0$), vectorized per-batch. |
| temperature $\tau$ | 0.1 | Temperature in the discriminator Softmax of Eq. 10; scales the magnitude of Dist before exponentiation so the Softmax stays in its informative regime regardless of environment dimensionality. Fixed across all six tasks. |
| noise_scale | 0.5 | Standard deviation for $x_0$'s noise component used when forming $x_t$ during $Dist$ estimation. |

*Table 3.* Performance across six environments. For navigation and manipulation tasks, we report average success rate (Avg Suc. Rate); For locomotion tasks, we report average return (Avg Return). Values are presented as mean (±standard deviation).

| | Navigation (Avg Suc. Rate) | | Manipulation (Avg Suc. Rate) | | Locomotion (Avg Return) | |
|---|---|---|---|---|---|---|
| Algorithm | (a) Ant-goal | (e) Maze2d | (b) Hand-rotate | (c) Fetch-pick | (d) Hopper | (f) Walker2d |
| DRAIL | 0.7142 (±0.0160) | 0.7780 (±0.0373) | 0.7775 (±0.2847) | 0.7052 (±0.3538) | 3182.60 (±85.25) | 3122.69 (±764.43) |
| GAIL | 0.6465 (±0.0542) | 0.6902 (±0.0826) | 0.9317 (±0.0541) | 0.2798 (±0.3316) | 2921.73 (±243.64) | 1698.25 (±411.42) |
| WAIL | 0.6127 (±0.0153) | 0.2978 (±0.0785) | 0.2370 (±0.3830) | 0.0000 (±0.0000) | 2609.28 (±814.05) | 1729.20 (±984.86) |
| VAIL | 0.7662 (±0.0365) | 0.6360 (±0.0382) | 0.5694 (±0.2960) | 0.8539 (±0.0551) | 2878.04 (±286.72) | 1156.52 (±221.67) |
| AIRL | 0.5467 (±0.0246) | 0.8239 (±0.0241) | 0.4595 (±0.1993) | 0.0000 (±0.0000) | 7.86 (±2.91) | -5.27 (±1.18) |
| DP | 0.8212 (±0.0135) | 0.5618 (±0.0268) | 0.9068 (±0.0136) | 0.8298 (±0.0127) | 1433.21 (±131.03) | 2204.41 (±226.28) |
| FP | **0.8334** (±0.0222) | 0.5420 (±0.0207) | 0.9032 (±0.0188) | 0.5460 (±0.0367) | 1950.41 (±170.32) | 2384.81 (±187.98) |
| FA-OPD (Ours) | 0.8225 (±0.0284) | **0.8731** (±0.0331) | **0.9794** (±0.0150) | **0.9984** (±0.0023) | **3358.95** (±72.31) | **4164.24** (±62.19) |

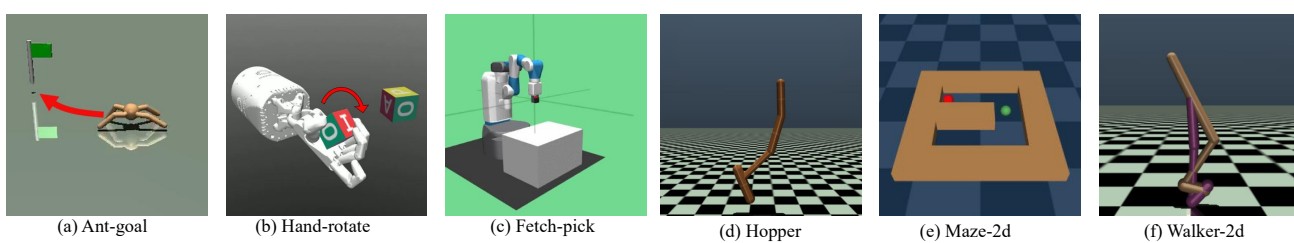

(a) Ant-goal    (b) Hand-rotate    (c) Fetch-pick    (d) Hopper    (e) Maze-2d    (f) Walker-2d

*Figure 5.* Overview of the six evaluation environments. *Navigation:* (a) **Ant-goal** tasks a quadruped agent with reaching a target position; (e) **Maze2d** requires an agent to navigate a 2D maze to a goal location; *Locomotion:* (d) **Hopper** requires fast and stable forward locomotion without falling; (f) **Walker2d** requires fast and stable forward locomotion without falling. *Manipulation:* (b) **Hand-rotate** requires dexterous in-hand rotation of a cube to a target orientation; (c) **Fetch-pick** requires grasping a block and placing it at a desired goal.

# F. Additional Experiment Results

## F.1. Quantitative Results

We provides the quantitative results of our main experiments in Table 3.

## F.2. Hyperparameter Study

The hyperparameter $\beta$ in Eq. 11 weights the action-distillation term against the reward-distillation term, and therefore controls the trade-off between the two distillation modes. As shown in Figure 6, we conducted an ablation study on $\beta$ in the Fetch-pick environment, testing values of $\{0, 0.1, 0.5, 1, 2\}$, where $\beta = 0$ ablates action distillation entirely (only reward distillation remains). The results show that $\beta \in \{0.5, 1, 2\}$ all converge faster and better than $\beta = 0$, confirming that action distillation adds value on top of reward distillation. Specifically, $\beta = 2$ achieves the best performance among the tested values; we use it for all main experiments.

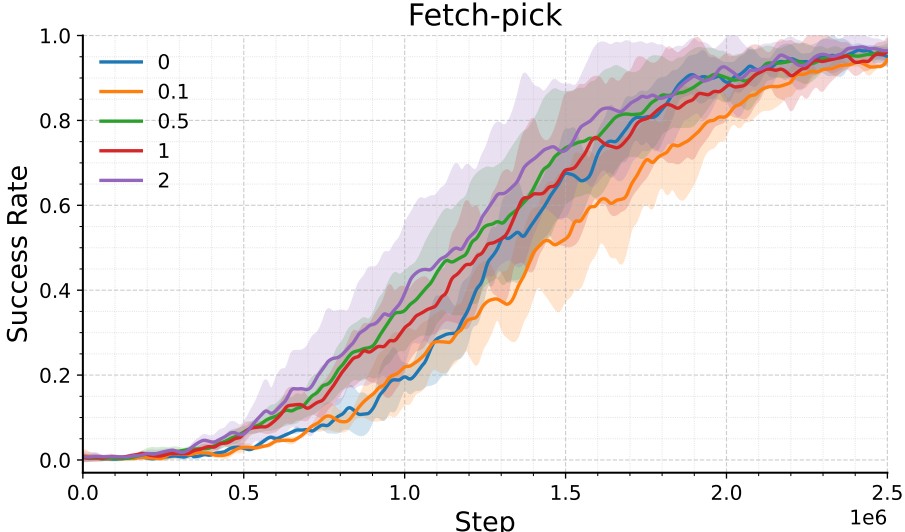

*Figure 6.* Performance of FA-OPD with different $\beta$ values in Fetch-pick environment.

**Is "larger $\beta$ always better"?** Figure 6 shows a monotonic ranking in the tested range $\beta \in \{0, 0.1, 0.5, 1, 2\}$, which could be read as "the heavier the action-distillation term, the better". To test this hypothesis we extend the sweep to $\beta \in \{5, 10\}$, where action distillation effectively dominates the reward-distillation signal. Each setting is run with two random seeds in Fetch-pick. As Table 4 reports, both larger values *slow* convergence relative to $\beta = 2$: $\beta = 5$ recovers to a comparable plateau by 7M steps while $\beta = 10$ remains noticeably behind. This contradicts the "larger is always better" interpretation and confirms the role of $\beta$ as a trade-off between the two distillation modes: when action distillation overwhelms reward distillation, the student behaves like a pure on-policy BC learner against the teacher and loses the online exploration that the reward term enables. The same intuition is reflected in the underperformance of the purely-imitative baselines DP and FP (Table 3).

*Table 4.* Extended $\beta$ ablation on Fetch-pick. Success rate at each training step, averaged over two seeds. Both $\beta = 5$ and $\beta = 10$ are slower to converge than $\beta = 2$ (the value used in our main experiments), showing that letting action distillation overwhelm reward distillation is harmful.

| Steps (M) | 0.5 | 1.0 | 1.5 | 2.0 | 2.5 | 3.0 | 3.5 | 4.0 | 4.5 | 5.0 | 5.5 | 6.0 | 7.0 |
|---|---|---|---|---|---|---|---|---|---|---|---|---|---|
| $\beta = 5$ | 0.00 | 0.00 | 0.01 | 0.01 | 0.03 | 0.03 | 0.29 | 0.77 | 0.94 | 0.94 | 0.96 | 0.99 | 0.99 |
| $\beta = 10$ | 0.00 | 0.00 | 0.00 | 0.01 | 0.02 | 0.01 | 0.03 | 0.17 | 0.56 | 0.84 | 0.97 | 0.94 | 0.97 |

## F.3. More Generalization Study

To further demonstrate the generalization ability of FA-OPD, we conduct more comprehensive experiments on the Navigation task Maze2d and the Locomotion task Walker2d. However, testing the agent's ability to generalize to unseen states requires different approaches for different task types. In Navigation tasks where the agent targets a specific location, applying noise to the goal position is not reasonable since it directly modifies the task objective itself, fundamentally changing what the agent is supposed to learn rather than testing generalization to state variations. Similarly, in Locomotion tasks where there are no explicit goal locations and the agent aims to maintain balance and move forward, the concept of goal noise is not

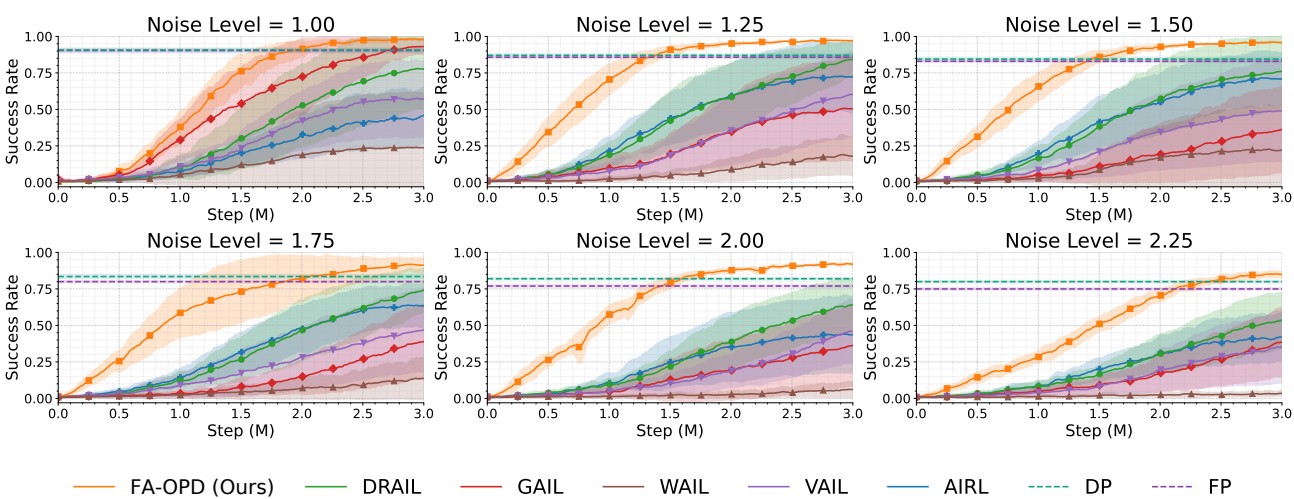

*Figure 7.* Learning curve of all methods in Hand-rotate environment across 6 noisy-levels.

*Table 5.* Performance of different algorithms across different number of expert demonstrations. For Maze environment, we report average success rate (Avg Suc. Rate) w.r.t four levels of expert data coverage: 25%, 50%, 75%, 100%; For Walker2d environment, we report average return (Avg Return) w.r.t four levels of expert $(s, a)$ transitions in datasets.

| | Navigation: Maze (Avg Suc. Rate w.r.t expert coverage (%)) | | | | Locomotion: Walker2d (Avg Return w.r.t # demo transitions) | | | |
|---|---|---|---|---|---|---|---|---|
| Algo. | 25 | 50 | 75 | 100 | 1k | 1.5k | 2k | 2.5k |
| AIRL | $0.8143_{\pm0.02}$ | $0.8528_{\pm0.02}$ | $0.9017_{\pm0.01}$ | $0.8729_{\pm0.02}$ | $-7.23_{\pm2.41}$ | $-2.51_{\pm1.89}$ | $-5.94_{\pm2.18}$ | $-4.62_{\pm2.03}$ |
| FP | $0.5138_{\pm0.02}$ | $0.6247_{\pm0.02}$ | $0.7316_{\pm0.04}$ | $0.7584_{\pm0.04}$ | $1421.58_{\pm118.74}$ | $1827.93_{\pm136.52}$ | $2084.71_{\pm161.38}$ | $2228.49_{\pm174.26}$ |
| DP | $0.5267_{\pm0.03}$ | $0.7419_{\pm0.03}$ | $0.8436_{\pm0.02}$ | $0.8917_{\pm0.02}$ | $1572.84_{\pm132.47}$ | $1896.25_{\pm149.83}$ | $2173.52_{\pm167.91}$ | $2468.37_{\pm189.15}$ |
| DRAIL | $0.8059_{\pm0.02}$ | $0.8516_{\pm0.02}$ | $0.9128_{\pm0.02}$ | $0.8642_{\pm0.02}$ | $1458.76_{\pm116.28}$ | $1691.84_{\pm143.16}$ | $2947.39_{\pm221.85}$ | $3066.72_{\pm241.93}$ |
| GAIL | $0.6534_{\pm0.04}$ | $0.6572_{\pm0.04}$ | $0.8127_{\pm0.03}$ | $0.8618_{\pm0.03}$ | $903.47_{\pm101.26}$ | $1158.65_{\pm128.39}$ | $1275.84_{\pm141.07}$ | $964.92_{\pm108.75}$ |
| VAIL | $0.5926_{\pm0.04}$ | $0.7438_{\pm0.03}$ | $0.8015_{\pm0.03}$ | $0.9024_{\pm0.02}$ | $691.35_{\pm90.18}$ | $679.47_{\pm85.43}$ | $697.28_{\pm91.86}$ | $967.51_{\pm116.02}$ |
| WAIL | $0.3184_{\pm0.05}$ | $0.1736_{\pm0.05}$ | $0.1652_{\pm0.05}$ | $0.5543_{\pm0.04}$ | $901.28_{\pm161.35}$ | $922.16_{\pm173.94}$ | $996.57_{\pm184.03}$ | $2457.85_{\pm322.16}$ |
| **FA-OPD (ours)** | $\mathbf{0.8273}_{\pm0.01}$ | $\mathbf{0.9231}_{\pm0.01}$ | $\mathbf{0.9426}_{\pm0.01}$ | $\mathbf{0.9718}_{\pm0.01}$ | $\mathbf{3026.81}_{\pm70.53}$ | $\mathbf{3635.42}_{\pm84.91}$ | $\mathbf{3812.68}_{\pm94.07}$ | $\mathbf{4368.93}_{\pm108.92}$ |

applicable. Hence, for Navigation and Locomotion tasks, we measure the generalization ability to unseen out-of-distribution (OOD) states by decreasing the coverage or amount of expert data. Specifically, we establish 4 expert coverage settings in Maze2d: 25%, 50%, 75%, and 100%, where k% means that the expert demonstrations cover k% of all possible scenarios in this environment. In Walker2d, we directly control the number of $(s, a)$ transitions in the expert data, creating 4 settings: 2500, 2000, 1500, and 1000 transitions. We then compare the final performance of each method under each setting, as shown in Table 5. Similar conclusions as in Section 4.2 could be derived based on these results.

### F.4. Controlled comparison of policy heads under a shared learned reward

To isolate the policy head from the reward signal, all methods here share the *same* learned reward from our FM-enhanced discriminator; the only varied factor is the policy architecture. We additionally include an **FM-PPO + AD** variant that applies the same action-distillation term of Eq. 11 to an FM policy, so an exact head-to-head with FA-OPD is possible. Table 6 reproduces the pattern of Sec. 4.4: all FM-policy variants fail to learn (action distillation only nudges FM-PPO from 0.13 to 0.16) while the MLP student reaches 0.88 under the same supervision, consistent with the gradient pathology analyzed in Appendix C.

## G. Baseline Details

In this section, we summarize the methodological foundations of all baselines used in our comparisons. When possible, we present their core objectives in mathematical form for clarity and reproducibility.

**Diffusion Policy (DP) (Chi et al., 2024).** DP models the conditional action distribution $\pi(a \mid s)$ via a conditional diffusion generative model. Let $\{q_t\}_{t=0}^{T}$ denote the forward noising process that gradually perturbs clean actions $a_0 \sim \pi_E(\cdot \mid s)$ into

*Table 6.* Controlled policy-head comparison on Maze2d. All rows share the same FM-enhanced learned reward; the only varied factor is the architecture and training of the policy head. The " + AD" rows additionally apply the action-distillation term of Eq. 11, ensuring an exact head-to-head comparison with FA-OPD.

| Algorithm (policy head) | Avg Success Rate |
|---|---|
| FM-A2C (FM policy) | $0.0950_{\pm 0.04}$ |
| FM-PPO (FM policy) | $0.1250_{\pm 0.08}$ |
| FPO (FM policy) | $0.0533_{\pm 0.01}$ |
| FM-PPO + AD (FM policy) | $0.1611_{\pm 0.050}$ |
| **FA-OPD (MLP policy + AD, ours)** | **$0.8750_{\pm 0.03}$** |

noisy variables $a_t$:

$$q(a_t \mid a_{t-1}) = \mathcal{N}\big(\sqrt{1 - \beta_t}\, a_{t-1},\, \beta_t I\big), \quad t = 1, \ldots, T. \tag{37}$$

A reverse-time parameterization $p_\theta$ denoises step-by-step conditioned on $s$:

$$p_\theta(a_{t-1} \mid a_t, s) = \mathcal{N}\big(\mu_\theta(a_t, s, t),\, \Sigma_\theta(a_t, s, t)\big). \tag{38}$$

Training minimizes the denoising score-matching loss (often in the $\epsilon$-prediction parameterization):

$$\mathcal{L}_{\mathrm{DP}}(\theta) = \mathbb{E}_{t,\, (s,a_0)\sim\mathcal{D}_E,\, \epsilon\sim\mathcal{N}(0,I)} \big\| \epsilon_\theta(a_t, s, t) - \epsilon \big\|^2, \tag{39}$$

where $a_t = \sqrt{\bar{\alpha}_t}\, a_0 + \sqrt{1 - \bar{\alpha}_t}\,\epsilon$ and $\bar{\alpha}_t = \prod_{i=1}^t (1 - \beta_i)$. Inference samples $a_T \sim \mathcal{N}(0, I)$ and iteratively applies $p_\theta$ to obtain $a_0$. DP is a supervise learning approach to clone the expert behavior.

**Flow-Matching Policy (FP) (Zhang et al., 2025a).** FP models $\pi(a \mid s)$ via conditional flow matching. Let $a_t$ follow an ODE driven by a conditional vector field $v_\theta$:

$$\frac{da_t}{dt} = v_\theta(a_t, s, t), \quad a_0 \sim \mathcal{N}(0, I), \quad a_1 \equiv a. \tag{40}$$

With a predefined conditional path $p_t(a \mid s, a_1)$ and teacher velocity $u_t(a_t \mid s, a_1)$, FP minimizes the FM regression:

$$\mathcal{L}_{\mathrm{FP}}(\theta) = \mathbb{E}_{t\sim\mathcal{U}[0,1],\, (s,a_1)\sim\mathcal{D}_E,\, a_t\sim p_t(\cdot|s,a_1)} \big\| v_\theta(a_t, s, t) - u_t(a_t \mid s, a_1) \big\|^2. \tag{41}$$

At test time, $a_1$ is obtained by numerically integrating the ODE from $a_0 \sim \mathcal{N}(0, I)$. FP is a supervise learning approach to clone the expert behavior.

**GAIL (Ho & Ermon, 2016).** GAIL frames imitation as matching occupancy measures via an adversarial game between policy $\pi_\phi$ and discriminator $D_\psi$:

$$\min_\psi \max_\phi\ \mathbb{E}_{(s,a)\sim\rho_{\pi_\phi}}\big[\log D_\psi(s,a)\big] + \mathbb{E}_{(s,a)\sim\rho_E}\big[\log(1 - D_\psi(s,a))\big] - \lambda\,\mathcal{H}(\pi_\phi), \tag{42}$$

where $\mathcal{H}$ is policy entropy regularization, $\rho_{\pi_\phi}$ is the state-action visitation distribution under $\pi_\phi$. The shaped reward for RL is $r(s,a) = -\log(1 - D_\psi(s,a))$.

**VAIL (Peng et al., 2018).** VAIL augments GAIL with an information bottleneck on the discriminator via a variational latent $z$ to reduce overfitting and improve robustness:

$$D_\psi(s,a) = \sigma\big(f_\psi(s,a,z)\big), \quad z \sim q_\psi(z \mid s,a), \tag{43}$$

and adds a KL constraint to enforce an information bottleneck:

$$\mathcal{L}_{\mathrm{IB}}(\psi) = \beta\,\mathbb{E}_{(s,a)}\big[\mathrm{KL}\big(q_\psi(z \mid s,a) \,\|\, p(z)\big)\big], \tag{44}$$

leading to the min-max:

$$\min_\psi \max_\phi\ \mathbb{E}_{\rho_{\pi_\phi}}\big[\log D_\psi(s,a)\big] + \mathbb{E}_{\rho_E}\big[\log(1 - D_\psi(s,a))\big] + \mathcal{L}_{\mathrm{IB}}(\psi) - \lambda\mathcal{H}(\pi_\phi). \tag{45}$$

**WAIL (Xiao et al., 2019).** Wasserstein Adversarial Imitation Learning replaces the JS divergence in GAIL with the 1-Wasserstein distance using a 1-Lipschitz critic $f_\psi$:

$$\max_{\psi \in \mathrm{Lip}(1)} \mathbb{E}_{\rho_{\pi_\phi}}\big[f_\psi(s,a)\big] - \mathbb{E}_{\rho_E}\big[f_\psi(s,a)\big], \tag{46}$$

with gradient penalty enforcing Lipschitzness:

$$\mathcal{L}_{\mathrm{GP}}(\psi) = \lambda_{\mathrm{gp}} \mathbb{E}_{\hat{x}}\big(\|\nabla_{\hat{x}} f_\psi(\hat{x})\|_2 - 1\big)^2, \quad \hat{x} = \epsilon x_E + (1-\epsilon)x_\pi. \tag{47}$$

The policy is trained with reward $r(s,a) = f_\psi(s,a)$.

**AIRL (Fu et al., 2017).** AIRL decomposes the discriminator to recover a reward function invariant to dynamics:

$$D_\psi(s,a,s') = \frac{\exp\big(f_\psi(s,a,s')\big)}{\exp\big(f_\psi(s,a,s')\big) + \pi_\phi(a\mid s)}, \quad f_\psi(s,a,s') = g_\psi(s,a) + \gamma h_\psi(s') - h_\psi(s), \tag{48}$$

where $g_\psi$ approximates the reward and $h_\psi$ the shaping potential. The implied reward for policy optimization is $r_\psi(s,a) = g_\psi(s,a)$.

**DRAIL (Lai et al., 2024).** DRAIL replaces the GAIL discriminator with a conditional diffusion model trained as a binary classifier via single-step denoising losses. For a state-action pair $(s,a)$ and condition $c \in \{c^+, c^-\}$ (expert vs. agent), define the class-conditional diffusion loss

$$\mathcal{L}_{\mathrm{diff}}(s,a,c) = \mathbb{E}_{t,\epsilon}\,\|\epsilon_\phi(s,a,\epsilon,t\mid c) - \epsilon\|_2^2, \tag{49}$$

approximated with a single sampled $(t,\epsilon)$. Let $\mathcal{L}_{\mathrm{diff}}^{\pm}(s,a) \equiv \mathcal{L}_{\mathrm{diff}}(s,a,c^{\pm})$. The diffusion discriminative classifier is

$$D_\phi(s,a) = \frac{e^{-\mathcal{L}_{\mathrm{diff}}^+(s,a)}}{e^{-\mathcal{L}_{\mathrm{diff}}^+(s,a)} + e^{-\mathcal{L}_{\mathrm{diff}}^-(s,a)}} = \sigma\big(\mathcal{L}_{\mathrm{diff}}^-(s,a) - \mathcal{L}_{\mathrm{diff}}^+(s,a)\big). \tag{50}$$

Train $D_\phi$ with BCE:

$$\mathcal{L}_D = \mathbb{E}_{\tau_E}[-\log D_\phi(s,a)] + \mathbb{E}_{\tau_i}[-\log(1 - D_\phi(s,a))]. \tag{51}$$

Policy optimization uses the adversarial (logit) reward

$$r_\phi(s,a) = \log D_\phi(s,a) - \log(1 - D_\phi(s,a)), \tag{52}$$

and any RL optimizer (e.g., PPO). This design avoids costly full diffusion sampling, yields a bounded, smooth "realness" signal, and aligns with GAIL's occupancy-matching objective via a class-conditioned diffusion discriminator.

**Implementation Notes.** All IRL baselines are trained with on-policy or off-policy RL updates using the shaped rewards defined above. Supervised baselines (DP and FP) are trained purely on $\mathcal{D}_E$ without online interaction, hence their evaluation curves are horizontal as they do not improve with additional environment steps.

### G.1. Comparison between FA-OPD (Ours) and DRAIL

Compared to DRAIL, FA-OPD introduces several key improvements: 1) **Enhanced discriminator efficiency via Flow Matching**: Our reward-distillation teacher uses an FM-based discriminator grounded in Optimal Transport, which requires fewer time discretization steps to generate high-quality state-action samples. DRAIL relies on a diffusion-based discriminator that demands multiple sampling steps to achieve comparable accuracy. 2) **Flexible probability paths**: The FM-enhanced discriminator supports a broader family of probability paths from noise to target state-action pairs. DRAIL is constrained to pre-defined diffusion processes, limiting representational capacity. 3) **Integrated action distillation**: FA-OPD additionally uses the same FM model as a generator to supply expert-distributed action targets at student-visited states (DAgger-style), at minimal computational cost. DRAIL has no analogous mechanism; the action-distillation term is essential for keeping the reward-distillation teacher's score reliable on the student's evolving on-policy distribution.

## H. Summary of Related Work

This section presents a comparative table summarizing the key characteristics of existing related works (Table 7). All listed works integrate generative models with decision-making, but they live in fundamentally different problem settings, which strongly determines what each can or cannot be compared against.

**Setting taxonomy.** We group the methods by their input requirements: *Offline RL* (DQL, IDQL, FQL) trains from a static $(s, a, r, s')$ dataset with no environment access; *Pre-trained fine-tuning* (DPPO, ReinFlow, One-Step FPMD) starts from a pre-trained FM policy and fine-tunes it with a known environment reward; *Online RL* (DIPO, SAC-Flow, QSM, Flow-GRPO) trains from scratch with a known environment reward; *Behavioral cloning* (Diffusion Policy, Flow Matching Policy) imitates an $(s, a)$ dataset with no reward and no online interaction; *Inverse RL* (GAIL family, DiffAIL, DRAIL, FA-OPD) recovers a reward from $(s, a)$ demonstrations and uses it for online policy optimization, *without ever observing the true environment reward*. Only the last group shares our setting and can be directly compared as a baseline; we therefore select Diffusion Policy, Flow Matching Policy, the GAIL family, and DRAIL as baselines. The *on-policy distillation* (OPD) literature (Agarwal et al., 2023; 2024) is a sibling framework conceptually adjacent to all of these but operates on language-model outputs with a pre-trained teacher, so it is omitted from the table for clarity; FA-OPD can be read as the natural extension of OPD to control with a learned, co-trained teacher.

**Why the IRL setting is the practically interesting one.** A common implicit assumption in much of the online-FM RL literature is that the environment reward is known. In many real-world deployment scenarios this is exactly the missing piece. Defining a hand-crafted reward for navigation in novel environments, dexterous manipulation, or driving is itself a hard research problem, and the data we can collect cheaply are demonstrations. This is exactly the regime that motivated us, and it is also where FA-OPD's gains are largest (see Sec. 4). Methods in the other settings are not competitors but complements: their advances in online FM optimization could in principle replace the PPO student inside FA-OPD while leaving the reward learner untouched.

*Table 7.* Comparison of policy learning frameworks across key dimensions. "Setting" indicates the problem input requirements: BC = behavioral cloning (only expert $(s, a)$); IRL = inverse RL (expert $(s, a)$, no true reward); Off. RL = offline RL from $(s, a, r, s')$; On. RL = online RL with a known reward; FT = pre-trained fine-tuning with a known reward. Only methods sharing our IRL setting are directly comparable as baselines; the rest are listed for context.

| Method | Setting | Online Expl. | Dist. Match | Stable Grad. | No Post-train | Efficient Inf. | Act. Distill. |
|---|---|---|---|---|---|---|---|
| Diffusion Policy | BC | | ✓ | ✓ | ✓ | ✓ | |
| Flow Matching Policy | BC | | ✓ | ✓ | ✓ | ✓ | |
| DQL / IDQL | Off. RL | | ✓ | | ✓ | | |
| FQL | Off. RL | | ✓ | ✓ | ✓ | ✓ | ✓ |
| DIPO | On. RL | ✓ | ✓ | ✓ | ✓ | | |
| QSM | On. RL | ✓ | ✓ | ✓ | | | |
| SAC-Flow | On. RL | ✓ | ✓ | ✓ | | ✓ | |
| Flow-GRPO | On. RL | ✓ | ✓ | | | | |
| DPPO | FT | ✓ | ✓ | | | | |
| ReinFlow | FT | ✓ | ✓ | | | ✓ | |
| One-Step FPMD | FT | ✓ | ✓ | ✓ | | ✓ | |
| GAIL family | IRL | ✓ | | ✓ | ✓ | ✓ | |
| DiffAIL / DRAIL | IRL | ✓ | ✓ | ✓ | ✓ | ✓ | |
| **FA-OPD (Ours)** | IRL | ✓ | ✓ | ✓ | ✓ | ✓ | ✓ |

Note: This table summarizes which capabilities are *explicitly supported* by each framework.
Key dimensions:
- *Online Expl.*: Supports active exploration through environmental interaction
- *Dist. Match*: Captures complex multi-modal distributions
- *Stable Grad.*: Avoids backpropagation instability
- *No Post-train*: Learns from scratch without fine-tuning
- *Efficient Inf.*: Enables low-latency policy execution
- *Act. Distill.*: Uses an action-distillation term (teacher-generated targets at student-visited states, DAgger-style (Ross et al., 2011)) in addition to the primary objective

# I. Statement of LLM usage

This paper employed LLM (specifically, Gemini3 Pro) to enhance the language quality of the writing. All content originated from the author, and the LLM was used solely to refine grammar, improve vocabulary usage, and enhance sentence-level coherence, without altering the original meaning.

