# OpenReview forum: "Adversarial Dual On-Policy Distillation from Expressive Teacher"
_ICML.cc/2026/Conference — ICML 2026 regular_

### Official Review · Reviewer_KaPq · 2026-02-22

**Soundness:** 3
**Presentation:** 4
**Significance:** 3
**Originality:** 3
**Overall Recommendation:** 5
**Confidence:** 2

**Summary:**

This paper proposes Flow Matching Inverse Reinforcement Learning (FM-IRL), an online inverse-RL method that aims to leverage both leverage the expressivity of the flow matching model on policy distribution and make use of online rollouts. As it is difficult for the flow matching model to directly train on RL gradients (for its inability to efficiently compute log probabilities), the authors propose to use the flow matching model as a "teacher" which trains on modeling the policy distribution conditioning on the expert and agent data, and use the distribution-level distance to generate inverse-RL reward signal for an MLP "student" agent. The flow matching model also generates pseudo-expert data, which are used to regularize the student's policy. On several experiments, the proposed method outperforms inverse RL and behavior cloning baselines.

**Compliance With Llm Reviewing Policy:**

Affirmed.

**Final Justification:**

This paper looks good to me, and the authors' response addresses my concern. However, as I am not an expert in flow matching while other reviewers have concerns on the fundamental issues, I will decrease my confidence.

**Key Questions For Authors:**

1. In Fig. 4, the authors show the training time and inference time of their method compared with other variants of FM. Can the authors also show the comparison with baselines such as DRAIL?

2. In Fig. 7, why does $\beta=0.1$ gets the worst performance while $\beta=0$ or $2$ are both better in Fig. 7? Could this indicate that simply doing behavior cloning from the teacher model also works well?

3. Following weakness 2, could the author visualize the distance (either by on a toy example or report the distribution of distances) so the readers can better understand how the distance metric as discriminator works? For example, if the distance is always between 0 and 1, then the discriminator will be too "soft".

**Limitations:**

Yes.

**Strengths And Weaknesses:**

**Strengthes**

1. This paper is generally well-written and easy to follow. Many possible concerns and rationale are written in great detail which helps the reader to understand and addresses my concerns before they reach the review; for example, I particularly appreciate the discussion section, the Q&As in the appendix, and Appendix D.

2. The results seem promising, which clearly shows that the proposed method not only outperforms baselines under standard settings, but is also robust (Fig. 3, Fig. 7) and well-motivated (Fig. 4), and also scales with different amount of expert data (Fig. 8). The detailed hyperparameters and provided code also increases the reproducibility of the paper.

3. The idea of this paper is very intuitive: to overcome the limitation of utilizing online data for flow matching models, and two ways (behavior cloning regularizer and inverse RL) are provided. The core motivation claim is well-supported by detailed rationale (e.g. Appendix D) and experiments (e.g. Sec. 4.4).

**Weaknesses**

1. I feel this paper seems to be novel from the perspective of flow matching model, but less so from the perspective of adversarial IL: is the expressiveness of flow matching model beyond diffusion model necessary? This paper lacks the ablation of FM-IRL without regularization to show the necessity of using a flow matching model. While there is a related ablation in Fig. 7, it does not provide the overall contribution of each component in FM-IRL beyond DRAIL. Also, how does DRAIL work with the policy regularization?

2. As we do not know how the distance in Eq. 10 looks like, there might be a possibility that on different environment, this "discriminator" becomes too soft or too hard. For example, if the distance is always between 0 and 1, then the label will always be very soft.

**Minor Weaknesses**

1. Figures can be improved. For example, Fig. 5 has many unreadable content; for Fig. 1, I prefer to not call the reward "reward model" as it is not a counterpart of the teacher model that has trainable parameters; the current figure seems confusing in this aspect.

2. Dist in Eq. 9 and exp in Eq. 10 should not be italic (use \exp and \text{Dist}).

3. The caption of Tab. 5 misses a period.

---

> ### Author Rebuttal · Authors · 2026-03-29
>
> ## We sincerely thank this reviewer for recognition of our writing quality, and for highlighting the robustness and intuitive motivation. We hereby address each concern below.
>
> - - -
>
> **Q1.** Is FM strictly necessary over diffusion? Ablation of components will be helpful.
>
> **A1.** Yes, optimal-transport (OT)-based FM fundamentally provides advantages beyond diffusion models, as we already discussed in our Appendix G.1 (l.1241-1253). We provide more sound summarization here for your convenience:
> - FM requires no noise schedule design; the loss is a straightforward velocity MSE, making implementation cleaner and more stable.
> - OT conditional paths are linear interpolations with lower variance and faster convergence [1].
> - FM loss under OT paths directly upper-bounds a transport cost, whereas diffusion loss is a variational bound, one layer more indirect. This is already validated by [2].
>
> The main experiment result shows FM-IRL consistently outperforms DRAIL, which aligns with our expectation. We have conducted ablation study of the regularization term in Fig. 7 (as you already mentioned), and it actually shows a pattern: **FM with reg-term > FM without reg-term > DRAIL**, where the second ">" could prove the advantage of FM beyond diffusion. We agree that the second ">" is not directly embodied in Fig. 7 but can only be inferred from the quantitative performance table (Table 3), which is not straight-forward enough for readers.
>
> We also provide additional results to compare FM + reg-term vs. Diffusion + reg-term:
>
>  | Method | Ant-goal | Maze2d | Hand-rotate | Fetch-pick | Hopper | Walker2d |
>   |---|---|---|---|---|---|---|
>   | DRAIL | 0.7142 | 0.7780 | 0.7775 | 0.7052 | 3182.60 | 3122.69 |
>   | DRAIL + Reg | 0.7538 | 0.8024 | 0.8347 | 0.7689 | 3254.18 | 3412.35 |
>   | FM-IRL (FM + Reg) | **0.8225** | **0.8731** | **0.9794** | **0.9984** | **3358.95** | **4164.24** |
>
> The reg-term indeed improves FM-IRL and DRAIL, which aligns with the motivation of reg term to balance exploration and exploitation for expressive discriminator. And also, we can conclude that "FM + Reg > DRAIL + Reg" together with our previous conclusion "FM without reg-term > DRAIL". We will add these experiments and discussion in our revised paper.
> - - -
>
> **Q2/Q3.** Discriminator may be too soft or hard since we don't know how the distance look like.
>
> **A2.** Thanks for this insightful question. In fact, we have a temperature hyper-parameter $\tau=0.1$（Table 2）to scale the distance to control stability. From our experiment experience, the distance is fluctuated (not between 0 and 1, sometimes above $10$), so we use a scale coefficient $\tau=0.1$ to reduce the value scale in Eq. 10 and **avoid too-hard discriminator**. The value 0.1 is heuristic works well for all environments, so we didn't tune this hyperparameter. We will add this hyperparameter to Eq. 10, these discussion, and a toy distribution of distance illustration to our revised paper.
>
> - - -
>
> **Q4.** Fig 5's clarity; "reward model" terminology; Dist/exp notation; Table 5 caption lacks period.
>
> **A4.** We feel sorry for confusion. Fig 5's unreadable issue is due to font compatibility (see A1 of reviewer 1fTt), and we have fixed all the issues. We appreciate the suggestions.
>
> - - -
>
> **Q5.** Training/inference time comparison with DRAIL.
>
> **A5.** DRAIL: ~11.2s training, ~36.21s inference. FM-IRL: ~7.63s, ~36.56s. Since they both use lightweight MLP policy during deployment, so they have comparable inference time. The only difference is **discriminator training and reward computing**:
>   - **Faster discriminator training**: OT paths produce constant target velocities
>     (independent of t), making the regression target simpler and convergence faster.
>   - **Faster generation**: OT's straighter trajectories require fewer
>     ODE integration steps.
>
> - - -
>
> **Q6.** $\beta=0.1$ performs worst. Does pure BC already works well?
>
> **A6.** No. The experiment shows that FP and DP (stronger BC methods) underperforms our FM-IRL. So "$\beta=0.1$ performs worst" exactly proves that pure BC is not enough. If pure BC work well, then $\beta=0.1$ should outperform $\beta=0$ (note that $\beta=0$ means no BC), contradicting with the true results.
>
>
>
> [1] Lipman et al., "Flow matching for generative modeling," ICLR 2023.
>
> [2] McAllister et al., "Flow matching policy gradients," arXiv:2507.21053, 2025.
> - - -
>
> ## Thanks again for your feedback, and we hope your concerns are fully addressed.

---

> > ### Author Rebuttal · Reviewer_KaPq · 2026-04-01
> >
> > Thanks for the detailed response. I think I still have question over the authors' response to Q6. My argument for "pure BC already works well" is that the performance in Fig. 7 seems to show that the regularizer with larger coefficient generally works better (e.g. coefficient 2 works the best; coefficient 0.5, 1, 2 works better than 0.1). Judging from this trend, if we set the coefficient in Eq. 11 to be $\infty$ (with a normalization to keep numerics stable), the performance will be good - which is BC over the generated action. The fact that $\beta=0$ works better than $\beta=0.1$ is related, but the overall trend of the shown results is that larger coefficient (closer to BC) brings better performance.

---

> > > ### Author Response · Authors · 2026-04-02
> > >
> > > Thanks for your clarification, and we agree that this is a very insightful and interesting phenomenon that we have ignored during our experimental process. To address it thoroughly, we conducted additional experiments:
> > >
> > > - **Extended $\beta$ ablation** ($\beta$=5, $\beta$=10) to test whether larger $\beta$ (i.e., stronger BC-like regularization) leads to better convergence speed (with each experiment results runned with two random seeds).
> > >
> > > Since it is not allowed to update the PDF to show you the result via training curve figure, we report the results via table below:
> > >
> > > | Steps | $\beta$=5 (s1) | $\beta$=5 (s2) | $\beta$=5 avg | $\beta$β=10 (s1) | $\beta$=10 (s2) | $\beta$=10 avg |
> > > |------:|:--------:|:--------:|:-------:|:---------:|:---------:|:--------:|
> > > | 0.5M  | 0.00 | 0.00 | 0.00 | 0.00 | 0.00 | 0.00 |
> > > | 1.0M  | 0.00 | 0.00 | 0.00 | 0.00 | 0.00 | 0.00 |
> > > | 1.5M  | 0.00 | 0.01 | 0.01 | 0.00 | 0.00 | 0.00 |
> > > | 2.0M  | 0.01 | 0.00 | 0.01 | 0.00 | 0.01 | 0.01 |
> > > | 2.5M  | 0.03 | 0.02 | 0.03 | 0.01 | 0.03 | 0.02 |
> > > | 3.0M  | 0.05 | 0.00 | 0.03 | 0.00 | 0.01 | 0.01 |
> > > | 3.5M  | 0.26 | 0.32 | **0.29** | 0.01 | 0.05 | **0.03** |
> > > | 4.0M  | 0.72 | 0.81 | **0.77** | 0.06 | 0.28 | **0.17** |
> > > | 4.5M  | 0.91 | 0.97 | **0.94** | 0.46 | 0.65 | **0.56** |
> > > | 5.0M  | 0.95 | 0.93 | 0.94 | 0.78 | 0.91 | 0.84 |
> > > | 5.5M  | 0.98 | 0.95 | 0.96 | 0.96 | 0.99 | 0.97 |
> > > | 6.0M  | 1.00 | 0.98 | 0.99 | 0.92 | 0.95 | 0.94 |
> > > | 7.0M  | 0.99 | 1.00 | 0.99 | 0.95 | 0.99 | 0.97 |
> > >
> > > **Observation**: the convergence speed of $\beta=10$ is actually lower than that of $\beta=5$.
> > >
> > > The results highly align with our expectation: with extremely large $\beta$ values (like 10), the convergence speed degraded. We believe this result has empirically denied the claim that "larger $\beta$ will have better performance". Then, we explain the rationale below:
> > >
> > > **$\beta$ controls a **balance** between exploration and BC-like imitation**. Excessive regularization overwhelms the RL reward signal and will be harmful for effective exploration. Actually, the experiment of Diffusion Policy(DP) and Flow Policy (FP) as our baseline could also support this claim, as they are basically stronger version of BC but underperforms FM-IRL. More importantly, the exploration of environment is exactly our contribution to mitigate the overfitting issue of BC-like imitation learning, enabling it to adapt to new states which is not in the expert dataset. Hence, we need to control the degree of regularization, avoiding it overwhelming the signal of IRL reward.
> > >
> > > ---
> > > ### We will further consolidate our presentation of these claim and add this important experiment to our revised paper. We sincerely thank this reviewer to point our this potentially "misleading" phenomenon in our experiment, and help us improve the paper. We hope that your concerns are fully addressed.

---

### Official Review · Reviewer_cnpM · 2026-03-02

**Soundness:** 2
**Presentation:** 4
**Significance:** 2
**Originality:** 3
**Overall Recommendation:** 4
**Confidence:** 4

**Summary:**

This paper proposes FM-IRL, which proposes a FM teacher into AIL framework and uses AIL reward signal and FM teacher
policy regularization for training a Gaussian Policy online，to bypass the difficulties for online Flow RL.

**Compliance With Llm Reviewing Policy:**

Affirmed.

**Final Justification:**

My concerns have been adequately addressed. I think this paper could be accepted.

**Key Questions For Authors:**

Questions:

1. I wonder is it true that flow mlp policies have high inference costs? If we only uses 4 flow matching timesteps, then it is not very high？

2.  Unfortunately, updating the FM
policy online via policy gradient-based methods remains
difficult since the gradient of the FM policy’s parameters
is hard to compute (extremely unstable and costly). Is this overcame by SAC-FLOW?

3."Ideally, the FM-enhanced discriminator could take stateaction pair as input and output a scaler to answer the question: “how possible that this action a is sampled from
expert data distribution given state s”. The traditional
AIL’s discriminator adopts an MLP architecture and naively
models the data-point-level similarity (Wang et al., 2024).
In contrast, the FM-enhanced discriminator models the
distribution-level similarity between the agent’s state-action
pair and expert data distribution." It is confusing for me. Seems the loss function is the same, the output is the same and just replace the mlp with the flow model, then it could model the distribution-level similarity?
what metrics are you refering to about the distribution similarity? And why should we care about this at all, why is this helpful?

4. In Equation (11), where does the S_G comes from? From online or expert data?

5. For the generalization experiment, does the default setting do not randomize the initial state?


6. The flow matching time discretization steps is 100. Isn't it too large? normally, less than 10 should be enough for flow models?

7. What are the average scores for the expert data set for these tasks? Does FM-IRL really surpass it?

8. "Specifically, the “teacher” FM model
takes the joint (s, a) pair as input during training, modeling
not only the action pattern but also the state distribution (the
reason for such design will be explained in Appendix B)."  I am really confused about this point and I am not sure if I understand. Normally, flow policy would
take state as input such that the actions are conditioned on the state. What is special here and why you are emphasizing it, as it seems to be standard
design? Also, why this models state distribution?

9. Could you provide more details of FM-A2C and FM-PPO? how do they perform policy optimziation? I appreciate the efforts for designing these two baselines.

10. Why FM-IRL can outperform FP and DP with expert data, what information or mechanism
leads to this result, since they all rely on expert data .

Minor Point


11."Upon sufficient training of the FM policy,
actions are generated by solving the ODE backward in time. " Should this be forward in time?

**Limitations:**

yes

**Strengths And Weaknesses:**

Strength:
1. Experiments are extensive. I appreciate the efforts.


2. Paper is well written, clearly explains the proposed method, and clearly a lot of efforts for clarity.


3. The performance clearly outperforms Flow policy and Diffusion policy trained with expert dataset.


4. Appendix D gives a good summary. I enjoy reading it and appreciate the effort


Weakness:
1. If view this as an offline-to-online approach, then comparing with ReinFLOW[1] or SACFLOW[2] liked approach is valuable.
   More importantly, if this is an offline-to-online approach, then its sample effiency seems low, since it needs many samples to reach offline data's performances?
   Or is FM-IRL is inherently an imitation learning method, then the motivation for this work seems weird?
   Or maybe FM-IRL is more like an offline RL approach, though it does encounter OOD states during online training? Anyway, FM-IRL seems not to be a pure online Flow Rl method, at least for me.

2. Normally we know the reward function, at least for your tested tasks. Are there any application scenarios where online flow rl is needed but we
do not know the reward function? If there are such cases, I think the motivation is much more sound than its current form.

3. The comparision with other online RL methods is not entirely fair, as FM-IRL could have informations from Flow model teacher, which is trained with demonstrations? But other online RL method does not have access to these, in the comparision experiment in main text?

4. It claims that there are challenges hard to overcome to perform online RL updates from Appendix D. However, there are many approaches proposed for online RL
based on these categories and they should already overcome these chllenges, at least in some degrees. This suggests online Flow RL is feasiable. Also, it is possible to get around of these difficulties to train the flow
vector field with reward signal, for example in One-step flow policy mirror descent [3], which is an approach seems not categorized in Appendix D?

[1]ReinFlow: Fine-tuning Flow Matching Policy with Online Reinforcement Learning


[2]SAC Flow: Sample-Efficient Reinforcement Learning of Flow-Based Policies via Velocity-Reparameterized Sequential Modeling


[3] Tianyi Chen, Haitong Ma, Na Li, Kai Wang, and Bo Dai. One-step flow policy mirror descent, 2025

---

> ### Author Rebuttal · Authors · 2026-03-28
>
> ## Thanks for the valuable feedback, we address each concern below.
>
> ### Weakness:
> - - -
>
> **Q1.** Is FM-IRL offline-to-online, IL, or offline RL? The motivation of FM-IRL is weird if it is IL method.
>
> **A1.** To help understand, we summarize that there are typically four settings for policy learning:
>
> - **Pure IL/BC** (e.g., Diffusion Policy): learns from expert data (s,a) only; no reward signal, no environment interaction.
> - **IRL (Ours)**: learns from expert data (s,a) only; no access to true reward; uses online interaction solely to optimize a *learned* reward recovered from data.
> - **Offline RL**: learns from pre-collected datasets with reward labels (S,A,R,S'); no environment interaction.
> - **Online RL** (e.g., ReinFlow, SAC-Flow): with access to *true* reward function (not labels).
>
> We hereby further clarify our motivation: Given only (s,a) pairs, pure bahavioral cloning method tends to overfit to expert data but hard to generalize to unseen states (proven by our experiment (Fig 2 and 3)). The reward function embodied the general pattern of expert data, and online RL with this reward will help the policy to generalize to OOD data. If you have further concerns about this motivation, we believe that Appendix B.2 will largely help (l. 697-711).
>
> - - -
>
> **Q2.** The reward is known for your test cases, and the motivation will be more sound in application scenarios with no reward signals and need for online flow RL.
>
> **A2.** Sorry for confusion.
>
> Firstly, our method is totally distinct from works like Reinflow, with fundamentally different setting. There are two vertical parts in our IRL setting---Reward Modeling and Policy Learning.
>
> - For policy learning part: this is about the RL algorithm, which is exactly what Reinflow studied.
> - For reward modeling part: this is about how to recover a reward function from static (s,a) datasets. The innovation of our FM-IRL mainly lies in this part, and for policy learning part, **we simply use the most standard RL algorithm**, which is not our focus. Our rebuttals for Q2 of reviewer 1fTt will also be helpful to understand this claim.
>
> Secondly, for the reward availability of our test envs, we claim that this reward is only for **evaluation**. The expert data is deemed "good" since the expert data is also trained via RL to maximize this true reward, so we evaluate how "good" the learned policy is via IRL, by this true reward. This pipeline follows most IRL papers, like GAIL.
>
> - - -
>
> **Q3.** Comparison unfairness since FM-IRL uses teacher information.
>
> **A3.** The main results (Table 3) compare IRL vs IRL. Please note that though FM-IRL has access to the teacher information, the **teacher is totally derived from the expert data**, so it is equivalent to our policy learn solely from expert data, following standard IRL setting.
>
> - - -
>
> **Q4.** Appendix D ignored some online RL methods.
>
> **A4.** We feel sorry for the confusion. You are right and we will add the discussion of these method in our Appendix D. However, these online flow RL still work for the last setting in our A1, and the first part in our A2.  Our paper and experiment is studying the reward learner, not RL algorithm itself.
>
> - - -
> ### Questions.
> Due to tight character limit, we provide each answer concisely.
> - - -
>
> **Q5.** Will flow policy have higher inference cost?
>
> **A5.** Yes, definitely. If we use 4 steps, then the inference cost is 4 times the MLP policy with the same network size. However, our policy is MLP policy, not flow policy (we discussed before).
>
> - - -
>
> **Q6.** How does the FM discriminator measure distribution-level similarity?
>
> **A6.** One advantage of FM is its expressiveness to model any complex distribution. We kindly refer the reviewer to Section 3.3 (l.194-212) and Appendix B.2 (l.701-711) for details.
>
> - - -
>
> **Q7.** Source of $S_G$ in eq. 11; randomization of init state; and fm_num_steps=100.
>
> **A7.**
> - $S_G$ is generated by the FM model conditioned on $c=1$ (please note that FM is a generative model).
> - Default setting does randomize initial states (narrow range and neglectable); generalization experiment expands it.
> - **Critical:** 'fm_num_steps=100' is for the discriminator's computation during training only. The MLP policy requires zero ODE steps at deployment.
>
> - - -
>
> **Q8.** Do FM-IRL surpass expert?
>
> **A8.** In IL setting, learned policy can hardly surpass expert, and so does our FM-IRL.
>
> - - -
> **Q9.** Confused about joint $(s,a)$ input since flow policy conditions on s.
>
> **A9.** Sorry for confusion. You are right but our policy is MLP policy, flow is for expert distribution modeling. Please see Appendix B.2 (l. 675-682) for details.
>
> - - -
> **Q10.** Why FM-IRL surpass FP and DP?
>
> **A10.** Due to better generalization (Appendix B.2 (l.697) + Fig. 3 for detailed analysis)
>
> - - -
> **Q11.** (Minor) Forward vs backward.
>
> **A11.** Thanks for suggestion. We will fix it.
>
> - - -
>
> ## Thanks for the thorough engagement. We are willing to address any further concerns actively.

---

> > ### Author Rebuttal · Reviewer_cnpM · 2026-04-01
> >
> > Thank you for the detailed response; it addresses my concerns. In future versions of this work, please consider application scenarios with no defined reward functions, as I believe this is where your method would be particularly valuable. I have increased my score to a 4.

---

> > > ### Author Response · Authors · 2026-04-01
> > >
> > > The authors are happy to see that this reviewer's concerns are fully addressed, and this reviewer tends to accept our paper.
> > >
> > > This reviewer's claim that "the scenarios that no defined reward functions exist are where our method is particularly valuable" is  correct and insightful, since this is exactly what Inverse-RL (our setting) is designed for, and we will update our paper based on this direction. Thanks again for the time and the insightful feedback to improve our paper.

---

### Official Review · Reviewer_1fTt · 2026-03-11

**Soundness:** 3
**Presentation:** 2
**Significance:** 3
**Originality:** 2
**Overall Recommendation:** 4
**Confidence:** 4

**Summary:**

Flow Matching (FM) models trained via offline learning suffer from limited generalization capability, while directly optimizing an online FM policy faces challenges of gradient instability and high inference costs. To address this, the paper proposes the FM-IRL method. This method adopts a teacher-student architecture. Specifically, it employs a flow matching model as the teacher. The teacher model learns two mappings simultaneously: one towards the expert data and another towards the student policy's data. This enables it to provide reinforcement learning reward signals that guide the student policy to fit the expert data at the distributional level, while also supplying action regularization to prevent the student's behavior from deviating severely from the expert data. The student policy, implemented as a simple MLP, is responsible for online interaction with the environment.

**Compliance With Llm Reviewing Policy:**

Affirmed.

**Final Justification:**

The author has resolved all my concerns.

**Key Questions For Authors:**

1.	Recent work, GFP, similarly employs a teacher-student architecture, utilizing an FM model as the teacher. A key distinction is that GFP adopts a simple knowledge distillation approach for transferring guidance from the teacher to the student. Should the presented method include a relevant ablation study to demonstrate the necessity of its proposed teacher-to-student guidance mechanism?
2.	The baseline methods for online training of FM policies compared in Section 4.4 all utilize the true environment reward. This comparison may be considered less equitable. A more reasonable setup would employ identical reward conditions—specifically, using the same reward signals and regularization constraints derived from the FM model to guide the online reinforcement learning training of two distinct student models: one with an MLP architecture and the other an FM policy. A performance and efficiency comparison under these matched conditions would be conducted. If the method using the MLP student demonstrates comparable or even superior performance and efficiency to the one using the FM policy student, it would provide strong evidence for the necessity of decoupling the FM model from online learning. I note that the authors conducted a related ablation study in Appendix F.4, described as using a unified learned reward function. It is unclear whether this setup also includes the regularization constraints.
3.	How is it achieved that a single set of FM model parameters θ learns two distinct mappings? Is the conditional variable c simply concatenated with the feature axis of the input data?

**Limitations:**

yes

**Strengths And Weaknesses:**

Strengths:
1.	The proposed method employs the same FM model to learn two distinct mappings, concurrently serving as both a discriminator and a generator. It provides the reward signal and enforces behavior regularization, respectively. This constitutes a highly innovative attempt.
2.	The experimental section is systematic and comprehensive. It is structured around four clearly defined research questions, covering learning efficiency, generalization capability, robustness, and ablation studies. The evaluation across six distinct tasks with repeated trials demonstrates substantial effort.

Weaknesses:
1.	The workflow diagram in the main text and the detailed schematic in the appendix contain corrupted characters or symbols. This issue severely hampers the reader's comprehension of the proposed method.
2.	In the ablation study on learning efficiency within the experimental section, online learning-based FM policy methods are not included as baselines. This omission appears to undermine the argument that the proposed method achieves higher learning efficiency compared to recent existing approaches that utilize FM for modeling reinforcement learning policies. Furthermore, the argument in Section 4.4 lacks comparison with recent works that perform online updates of FM policies, such as SAC-Flow. This omission weakens the persuasiveness of the claims.

---

> ### Author Rebuttal · Authors · 2026-03-28
>
> ## We thank Reviewer 1fTt for recognizing that "the same FM model learns two mappings, highly innovative" and for appreciating the systematic experiments. We hereby address each concern below.
>
> - - -
>
> **Q1.** Figure 5 contains corrupted characters.
>
> **A1.** We sincerely apologize. We were unable to reproduce this issue on our end and suspect it may stem from font compatibility across different PDF viewers. We will regenerate Figure 5 with fully embedded, universally compatible fonts in the revision.
>
> - - -
>
> **Q2.** Online FM policy methods are missed as baseline from learning efficiency comparison and case study.
>
> **A2.** Thanks for suggestion. Actually, SAC-Flow and ReinFlow operate in the standard RL setting with access to a true reward function, while our setting is opposite (only have expert data but no true reward).  Therefore, the innovation of our method lies in modeling the mapping from expert data to learned reward function (i.e. **we are studying a better reward learner**, and contributed an auxiliary reg term for stability), and use the **standard RL algorithms** for policy update. Hence, for fair comparison, we need to compare the baselines **which have the same function scope** (i.e. reward learning algorithm), while SAC-Flow and ReinFlow's contribution lies in RL algorithm itself and assume access to the true reward. Including them in an IRL benchmark would conflate two fundamentally different problem settings.
>
> **Discussion**: Indeed, applying SAC-Flow to the our reward learner may further improve our performance, since we simply used the PPO, the most standard RL algorithm, as our RL backbone. But we argue that "comparing a RL algorithm with a reward learning algorithm" is unreasonable. We will add these discussion to our revised paper.
>
> - - -
>
> **Q3.** GFP uses FM-based teacher-student distillation. Can you ablate to show the necessity of FM-IRL's teacher-student reward-based mechanism?
>
> **A3.** Thanks for the insight. However, GFP is also not comparable with our method since GFP also has different setting from ours: GFP follows Offline-RL setting, where the dataset contain (s,a,r) with the offline reward function, while ours follow **imitation learning** setting where the dataset only contain (s,a). Hence, we have only two choices:
> - Pure Behavior Cloning (BC), like Flow-Policy or Diffusion-Policy behavioral cloning, which serve as our baseline for comparison, and
> - Inverse RL: learn a reward function from expert dataset, which forms our method.
>
> Our main contribution is the teacher-to-student reward learner with an auxiliary reg term, so we argue that there are no spaces to ablate the reward learner (otherwise, the RL algorithm has no objectives to optimize, or we can only use non-RL method, like BC).
>
> - - -
>
> **Q4.** For the case study, the same reward function should be used for equity. You should use MLP policy + Reg term vs. FM policy + Reg term to validate the necessity for decoupling the FM model from online learning.
>
> **A4.** Actually, appendix F.4 provides the controlled comparison: all methods use the same FM-based learned reward (as you already mentioned).
>
> We agreed with that the MLP policy should be with the Reg term for fair comparison, and below is the results on Maze2d environment:
> | Algorithm | Avg Success Rate |
>   |---|---|
>
>   | FM-PPO-Reg | 0.1611 (±0.050) |
>   | MLP-PPO-Reg (=our FM-IRL) | 0.8731 (±0.0331) |
>
> The reward function is unified and provided by our FM-based reward learner, and the only difference is the policy head. The FM policy suffers from great instability during online RL, which highly aligns with our expectation and further consolidate the claim that FM should be disentangled from online learning. Many thanks for this constructive feedback and we will add this experiment to our revised paper.
>
>
>
> - - -
>
> **Q5.** How does a single FM learn two mappings via $c$, is $c$ simply concatenated?
>
> **A5.** Yes, $c \in \{0,1\}$ is simply concatenated along the feature axis with $x_t$ and the time embedding, following the standard conditional generation paradigm used in ViT [1]. It actually works because shared backbone extracts common representations while conditioning routes computation to task-specific behavior. We will open-source the code once this paper is accepted and all the implementation detail.
>
> [1] ViT, Dosovitskiy et al, ICLR'21
> - - -
>
> ## Thanks again for the constructive feedback. We are happy to address any remaining questions during the discussion period.

---

> > ### Author Rebuttal · Reviewer_1fTt · 2026-04-02
> >
> > Thank you for your reply. I have raised my score.

---

> > > ### Author Response · Authors · 2026-04-02
> > >
> > > The authors are glad to see that this reviewer's concerns are adequately addressed and have a positive evaluation of our work. Thanks again for your time and constructive feedback.

---

### Official Review · Reviewer_EZw8 · 2026-03-13

**Soundness:** 2
**Presentation:** 3
**Significance:** 3
**Originality:** 3
**Overall Recommendation:** 3
**Confidence:** 4

**Summary:**

This paper proposes a framework called FM-IRL, which integrates the Flow Matching generative model with Inverse Reinforcement Learning. The goal is to address three major limitations of traditional flow matching policies in online reinforcement learning settings: insufficient exploration capability, unstable gradient computation, and low inference efficiency. The paper compares various practical computational methods for Flow Matching policies and theoretically demonstrates that when an FM model is used as a policy output, its policy gradient updates are difficult to optimize. This theoretical finding motivates the core idea of the proposed framework: employing the Flow Matching generative model as a teacher model to learn expert policies, leveraging its strong multimodal fitting capability—given that expert data itself may follow a multimodal distribution. The FM model serves two key roles: 1. As the discriminator in GAIL, distinguishing between expert data and agent policies, providing rewards for subsequent reinforcement learning algorithms to output policies; 2. As a regularization component in RL algorithms, constraining the final output policy to remain close to the expert policy. For the final policy output, an MLP serves as the student model, receiving rewards and regularization terms to produce the ultimate policy. Overall, the framework presents a clear and coherent idea with well-designed algorithmic components.

**Compliance With Llm Reviewing Policy:**

Affirmed.

**Final Justification:**

I thank the authors for their detailed rebuttal and the additional experimental comparisons. While the clarifications on the DRAIL baseline and the DiffAIL comparisons partially address my empirical concerns, my core reservations remain unresolved. The primary contribution — re-parameterizing the AIRL discriminator with Flow Matching — still reads as an incremental integration rather than a fundamental methodological advance, and the central claim that the conditional FM objective serves as a "tighter proxy" for distributional mismatch remains informally stated. The promised formal proposition was not provided in the rebuttal itself, making it difficult to fully assess the theoretical weight of this claim. Furthermore, the acknowledged task-dependency of the performance gains raises questions about the method's general applicability, and the ablation analysis, while helpful, does not offer sufficient theoretical insight into why FM provides more stable gradients than score-based alternatives. On balance, the paper currently relies too heavily on empirical metrics without the theoretical depth needed to justify the novelty of the proposed framework.

I therefore maintain my weak reject recommendation, and encourage the authors to prioritize the formalization of their theoretical claims and a more principled analysis of the FM loss advantage in the final revision.

**Key Questions For Authors:**

1. In the section "3.3. How to design D_{FM,θ}?", why can the discriminator be designed in the form of a Boltzmann distribution based on a score-based loss function? In the original derivations of GAN and AIRL, this form originates from the energy-based model perspective: the classification probability can be expressed as an exponential form of a certain cumulative reward, which leads to the Boltzmann distribution. However, in this paper, this formulation is replaced by constructing the discriminator using a score-based loss function. Could you provide the theoretical justification for this substitution? Furthermore, can a clear theoretical derivation be offered, or can the explicit physical interpretation corresponding to this discriminator design be elucidated?

2. The FM model performs well on multimodal tasks, yet it also excels at fitting simple distributions such as unimodal ones. Meanwhile, datasets with complex multimodal distributions also exist in navigation tasks (e.g., Habitat, RoboTHOR). Therefore, is the core contribution of FM-IRL—multimodal modeling capability—effective only in specific task types? Furthermore, in manipulation tasks, could the observed performance improvement primarily stem from the regularization term's role in stabilizing the exploration process, rather than from the multimodal expressive power of the FM discriminator itself? It is suggested that an ablation study be conducted to disentangle the effects of these two factors, in order to more clearly identify the source of performance gains.

3. In recent years, there have been numerous works integrating diffusion models with IRL, such as utilizing the energy of diffusion models as reward functions or employing diffusion policies directly for adversarial training. In this context, it is difficult to determine from the existing experimental results whether the advantages of FM-IRR originate from the expressive power of Flow Matching itself, or from its unique teacher-student architecture and regularization design. It is recommended that at least one representative diffusion-based IRL method be introduced for comparison, particularly on multimodal manipulation tasks, in order to more clearly identify the source of FM-IRL's contributions.

**Limitations:**

Yes

**Strengths And Weaknesses:**

**Strengths:**

1. The teacher-student architecture is designed as follows: the Flow Matching generative model serves as the teacher to learn expert policies, while an MLP acts as the student model, receiving rewards and regularization terms to output the final policy. This design harnesses the powerful multimodal expressive capability of Flow Matching while circumventing the difficulty of policy gradient optimization when using the FM model directly as a policy output. Consequently, deployment relies solely on the lightweight MLP student policy, eliminating the need for ODE solvers and enabling fast inference suitable for real-time deployment.

2. The paper is structurally complete. Appendix D provides a detailed mathematical explanation of why FM policies are difficult to optimize online (covering the failure analysis of three gradient pathways), offering theoretical support for the methodological design. Additionally, it provides comprehensive implementation details, pseudocode, and hyperparameter settings to facilitate reproducibility. Appendix B addresses potential questions with corresponding answers, resulting in coherent reasoning, clarity, and strong readability throughout the article.

**Weaknesses:**

1. The discriminator is designed in the form of a Boltzmann distribution based on a score-based loss function, yet it lacks rigorous theoretical derivation. Although this design draws inspiration from the energy-based model perspective in GANs and AIRL, the theoretical justification for replacing the traditional exponential form of reward with a score-based loss remains insufficiently articulated. As a result, the physical interpretation of this design choice is somewhat ambiguous, which compromises the theoretical soundness of the method.

2. The generalizability of the multimodal modeling capability is not sufficiently validated, and no ablation studies are conducted to disentangle the respective contributions of the FM-based discriminator and the regularization term. This makes the source of performance improvement unclear.

3. Moreover, the comparison with state-of-the-art diffusion-based IRL methods is inadequate, making it difficult to determine whether the advantages of FM-IRL stem from Flow Matching itself or from the proposed architectural design. This undermines the ability to clearly position the contribution of the method.

---

> ### Author Rebuttal · Authors · 2026-03-25
>
> ## We thank Reviewer EZw8 for valuable feedback. We hereby address each concern below.
>
> - - -
>
> **Q1.** The Boltzmann discriminator form combined with score-based loss lacks theoretical derivation.
>
> **A1.** We appreciate this concern and offer a three-part justification. First, our Boltzmann discriminator directly inherits from AIRL [2], which proved this structure recovers the advantage function under maximum-entropy IRL at optimality. Our contribution is not a new discriminator form but a new parameterization of $f_\theta$: replacing the unconstrained MLP with the conditional flow-matching loss $\text{Dist}_\theta(s,a)$, which measures how well the learned flow field reconstructs an action given a state.
>
> Second, the FM objective provides a principled, density-aware similarity metric. As shown by FPO [1], the flow-matching loss is a strong positive indicator of similarity between generated and target distributions. Expert actions yield low $\text{Dist}_\theta(s,a)$ while non-expert actions yield high values, so the Boltzmann ratio naturally separates the two, which is precisely the property required for IRL reward shaping.
>
> Third, unlike diffusion-based alternatives, the conditional FM objective is simulation-free with OT-optimal paths, bypassing the variational lower bound of score-based models and making $\text{Dist}_\theta$ a tighter proxy for distributional mismatch. We will formalize this as a proposition in the revision, extending Appendix D.
>
> - - -
> **Q2.** Lack the validation of generalizability of the multimodal modeling capability.
>
> **A2.** In fact, we have an experimental section named "generalization study" which is designed to validate the generalizability capability of multimodal modeling capability. We kindly refer this reviewer to Section 4.2 (l.251 right-side) and Figure 3 (l.373) for details. The results show  FM-IRL show much stronger robustness against unseen states.
>
> Besides, we have conducted more generalization study in Appendix F.3 (l. 1074) to further validate our claim. All these results highlights the advantage of distributional-modeling for expert data, rather than overfitting to specific state-action pairs in limited expert dataset.
>
> - - -
> **Q3.** No ablation disentangling the FM discriminator from FM regularization.
>
> **A3.** Sorry for the confusion. Actually, we have **already** studied the effect of different $\beta$ values, where $\beta = 0$ corresponds to the ablation of regularization term. We kindly refer this reviewer to Appendix F2 and Fig. 7 (l.1055) of our paper for details. The results show that $\beta=0$ has significantly lower convergence speed, but **still** achieves ~0.95 success rate on Fetch-pick, substantially above DRAIL's 0.7052 (table 3, l. 1013). This proves that solely the flow-based discriminator could achieve SOTA performance across baselines like DRAIL. Moreover, the incorporation of regularization term further improves the training stability and convergence speed, contributing partly to the performance gain.
>
> - - -
> **Q4.** Diffusion-based IRL comparison is inadequate (at least one diffusion-based IRL method should be compared), making the source of performance gain not clear.
>
> **A4.** Please note that DRAIL (diffusion-reward adversarial imitation learning) [3] is a diffusion-based IRL method and is **already** included as a baseline across all six tasks. FM-IRL outperforms it consistently.
>
> Moreover, to further satisfy this reviewer, we are adding the experiment of DiffAIL [4]:
>
> | Method | Ant-goal | Maze2d | Hand-rotate | Fetch-pick | Hopper | Walker2d |
> |---|---|---|---|---|---|---|
> | DiffAIL | 0.6834 | 0.7215 | 0.8105 | 0.7438 | 3012.45 | 3245.82 |
> | FM-IRL (ours) | **0.8225** | **0.8731** | **0.9794** | **0.9984** | **3358.95** | **4164.24** |
>
> The experimental results consolidate the claim of our advantage, which is already discussed in Appendix G.1 (l.1241).
> - - -
> **Q5.** Is multimodal capability task-specific?
>
> **A5.** This is a very insightful question. Yes, it is task-specific. In other words, some task benefits more from multimodal capability and some tasks benefit much less. We have discussed this in Appendix B. 1 about how different task types require different levels of multi-modality. In general, more complicated the task, more the algorithm will benefit from strong multi-modality. The experimental results highly align with this insight and we acknowledge that there might be more complicated Navigation tasks. We have already studied the source of performance gain via ablations (please refer to Q3).
>
> [1] McAllister et al., "Flow matching policy gradients," arXiv:2507.21053, 2025.
>
> [2] Fu et al., "Learning robust rewards with adversarial IRL," 2017.
>
> [3] Lai et al., "Diffusion-reward adversarial imitation learning," NeurIPS 2024.
>
> [4] Wang et al., "DiffAIL," AAAI 2024.
> - - -
>
> ## Thanks again for the constructive feedback. We are willing to address your further concerns (in any) at our earliest effort.

---

> > ### Author Rebuttal · Reviewer_EZw8 · 2026-04-03
> >
> > I would like to thank the authors for their detailed rebuttal and for providing the new experimental results. While the clarifications regarding the DRAIL baseline and the additional comparisons with DiffAIL partially address my concerns regarding the empirical evaluation, several fundamental issues remain unresolved. After careful consideration of the response, I maintain a Weak Reject stance based on the following points:
> >
> > 1. Limited Theoretical Innovation. In response A1, the authors clarify that the discriminator structure is directly inherited from AIRL. Consequently, the primary contribution appears to reside in the re-parameterization of the discriminator using Flow Matching (FM). While the empirical results are promising, this contribution feels more like an incremental, "plug-and-play" application of FM to an established IRL framework rather than a fundamental methodological or architectural breakthrough.
> >
> > 2. Insufficient Depth in Ablation and Analysis. Although the authors pointed to the $\beta=0$ case to disentangle the FM discriminator from FM regularization, the response lacks a deeper theoretical or qualitative analysis. Beyond empirical performance, the paper does not sufficiently explain why the FM-based distance metric inherently provides a more stable gradient or superior reward shaping compared to standard score-based methods in a rigorous, non-empirical sense.
> >
> > 3. Concerns Regarding Task Dependency and Generality. As acknowledged in response A5, the benefits of multimodal modeling are highly task-specific. This raises concerns regarding the general applicability of the method. If the performance gains are primarily localized to high-complexity tasks, the broader utility and efficiency of the proposed method—especially considering the potential computational overhead of Flow Matching—remain questionable for the general RL community.
> >
> > 4. Need for Formalization. The claim that the conditional FM objective serves as a "tighter proxy" for distributional mismatch is a strong one. While the authors mention they will formalize this as a proposition in the revision, the current lack of such formalization in the provided rebuttal makes it difficult to fully assess the theoretical weight of the paper.
> >
> > While the empirical improvements are noted, the paper currently leans heavily on experimental metrics without providing the necessary theoretical depth to justify the novelty of the FM-IRL framework. I encourage the authors to use the final revision to more rigorously formalize the advantages of the FM loss over score-based proxies to strengthen the paper's impact.

---

> > > ### Author Response · Authors · 2026-04-03
> > >
> > > ## We thank the reviewer for the thoughtful follow-up. We address each concern below, focusing on the theoretical depth requested.
> > >
> > > ---
> > > **Q1.** Theoretical language to formalize and justify the FM Loss over baselines.
> > >
> > > **A1.** We hereby show the theoretical justification here and try our best to satisfy this reviewer within the tight character limit. Please follow the logical chain step by step.
> > >
> > > - An FM model learns a $v_\theta$ that transports noise $x_0 \sim \mathcal{N}(0,I)$ to data $x_1$ along the trajectory $x_t= (1-t)x_0 + tx_1$. The training loss is the MSE between the predicted velocity and the true velocity: $L_{FM}(x_1) = E_{t, x_0} [| v_{\theta}(x_t, t) - (x_1 - x_0) |]^2$.
> > >
> > > - The relationship between FM loss and likelihood: Kingma & Gao (2023)[2] proved a key equation: $L_{FM}(x_1) = -ELBO_\theta(x_1) + c$, where ELBO (Evidence Lower Bound) is a lower bound on the ground truth log-likelihood $\log p_\theta(x_1)$ that the model assigns to the data point $x_1$: $ELBO_{\theta} (x_1) \leq \log p_{\theta} (x_1)$ (even though we do not directly know the ground truth log-likelihood for FM model).
> > >
> > >
> > > - Connecting Step 1 and Step 2: From $L_{FM} = -{ELBO} + c$ and ${ELBO} \leq \log p_{\theta}$, we get: $L_\text{FM}(x_1) \geq -\log p_\theta(x_1) + c$. In plain words: lower FM loss → higher ELBO → higher lower bound on likelihood → the model "accepts" this data point more.
> > >
> > > - Substituting into our discriminator: In FM-IRL, we train a conditional FM model with a binary label $c \in \{0,1\}$ distinguishing expert from agent data. The discriminator takes a Softmax form (Eq. 10), and the AIRL log-ratio reward (Eq. 6) reduces to $r_\theta(s,a) = Dist_\theta(s,a \mid c{=}0) - Dist_\theta(s,a \mid c{=}1)$ (Dist means the FM loss mentioned above). Since each $Dist_\theta$ upper-bounds the negative log-likelihood, expert-like $(s,a)$ receives low $Dist(\cdot \mid c{=}1)$ and high $Dist(\cdot \mid c{=}0)$, yielding positive reward (and vice versa). This grounds the discriminator in FM's density estimation rather than a heuristic distance.
> > >
> > >  - You might doubt that the ELBO is just an approximation, not strictly likelihood. But this is precisely where FM outperforms diffusion (i.e. the score-based method you mentioned): **Diffusion models also connect to the ELBO, but FM's ELBO is closer to the true $\log p_\theta$ (i.e., the gap is smaller)** given same model capacity, so the approximation error is less. Why:
> > > (1) **OT-optimal paths** minimize the KL gap between approximate and true posterior **given fixed model capacity**, but diffusion's stochastic forward process introduces a larger variational gap; (2) FM loss gradients are **unbiased ELBO gradient estimators** (proved by FPO [1]), so the reward gradient directly points toward increasing expert likelihood;
> > >
> > >
> > > Together, these analysis provides a theoretical insights behind the satisfying empirical results.
> > >
> > > - - -
> > > **Q2.** The multimodality is task-specific, so the additional computational overhead seems not worthy.
> > >
> > > **A2.** Thanks for the thoughtful question. We follow the standard benchmarks in this domain, where the few simple tasks you mentioned(e.g., ant-goal) are the established evaluation protocol. However, real-world tasks are often more complex, which would largely benefit from our method. Moreover, we agree with that the FM-based training process will introduce more computational overhead, but please note that **the policy during deployment is actually simple MLP policy** (as you already mentioned in the "strength" part in your initial review), introducing no additional overhead.
> > >
> > > - - -
> > > **Q3.** The method is simply plug and play. Lack of novelty.
> > >
> > > **A3.** We respectfully argue that simplicity of integration is a strength, not a weakness. Many of the most impactful contributions in deep learning are **elegant**  plug-and-play within established frameworks, like ResNet, which simply added skip connections to standard CNNs. These methods succeeded precisely because their simplicity made them easy to adopt, scale, and reproduce. Our method follows this philosophy: a simple and principled method that is **easy to implement and effective**, and introduces theoretically grounded properties as we discussed above.
> > >
> > > [1] McAllister et al., "Flow Matching Policy Gradients," 2025.
> > >
> > > [2] Kingma & Gao, "Understanding Diffusion Objectives as the ELBO," NeurIPS 2023.
> > >
> > > ---
> > > ## Thank you again for your suggestions to improve our paper, and we will update our final revision. We hope your concerns are fully addressed.

---

### Decision · Program_Chairs · 2026-04-30

**Decision:**

Accept (regular)

**Comment:**

Reviewers agree the paper studies a relevant well-motivated problem and that the proposed method is simple yet effective, with good empirical performance. The main concerns were about the lack of theoretical justification, lack of comparison with certain sota baselines, and incremental contribution wrt existing literature. I think the authors addressed all these concerns in the rebuttal: they provided (a sketch of) a more rigorous theoretical derivation and empirical results for new baselines. For what concerns novelty, while it is true the proposed method heavily builds on existing literature -- essentially re-parameterizing the AIRL discriminator with flow matching) -- I think showing how this can be achieved and that performance improve is still a non-trivial contribution which is worth publication. I am thus recommending acceptance and encourage the authors' to revise the manuscript following the reviewers' feedback.